# LASER: Learning Active Sensing for Continuum Field Reconstruction

**Huayu Deng** [1]  **Jinghui Zhong** [1]  **Xiangming Zhu** [1]  **Yunbo Wang** [1]  **Xiaokang Yang** [1]

## Abstract

High-fidelity measurements of continuum physical fields are essential for scientific discovery and engineering design but remain challenging under sparse and constrained sensing. Conventional reconstruction methods typically rely on fixed sensor layouts, which cannot adapt to evolving physical states. We propose *LASER*, a unified, closed-loop framework that formulates active sensing as a Partially Observable Markov Decision Process (POMDP). At its core, LASER employs a **continuum field latent world model** that captures the underlying physical dynamics and provides intrinsic reward feedback. This enables a reinforcement learning policy to simulate "what-if" sensing scenarios within a latent imagination space. By conditioning sensor movements on predicted latent states, LASER navigates toward potentially high-information regions beyond current observations. Our experiments demonstrate that LASER consistently outperforms static and offline-optimized strategies, achieving high-fidelity reconstruction under sparsity across diverse continuum fields.

## 1. Introduction

Achieving high-fidelity observation of continuum physical fields, such as fluid flows, stress-strain distributions, or spatiotemporal temperature fields, is fundamental to scientific discovery and engineering systems, with broad impact on climate modeling, material analysis, and industrial control (Evans, 2022; Brunton & Kutz, 2022; Takamoto et al., 2022). In practice, however, physical, economic, and environmental constraints often restrict sensing to a sparse and discrete set of spatial locations, making the full field only partially observable (Huang et al., 2024; Serrano et al., 2023; 2024; Luo et al., 2024; Ma et al., 2025). This intrinsic

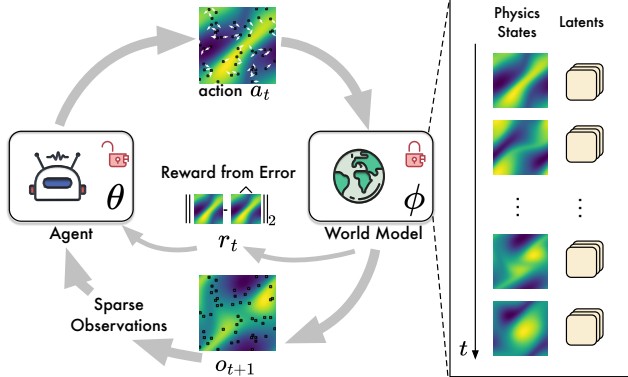

*Figure 1.* **Overview of the closed-loop LASER framework.** The agent $\theta$ optimizes sensing actions $a_t$ based on reconstruction rewards $r_t$ and next-step observations $o_{t+1}$ from the environment $\phi$. The side panel illustrates the temporal evolution of high-fidelity physical states and their corresponding low-dimensional latents.

mismatch between the continuous nature of physical dynamics and the sparsity of discrete observations creates a twofold challenge: One must not only reconstruct the global field from limited data (Koupaï et al., 2025; Serrano et al., 2024; Alkin et al., 2024; Serrano et al., 2023) but also solve the inverse problem of **optimal sensor placement** to ensure the most informative regions are captured.

Empirically, sensor placement is a primary determinant of reconstruction accuracy. Even with an identical reconstruction model, varying the sensing layout can lead to dramatically different results (Ma et al., 2025). While contemporary methods explore the joint optimization of sensing and reconstruction, they predominantly rely on decoupled, offline paradigms whereby sensor locations are optimized over a global training distribution but remain static during inference (Ma et al., 2025). These static layouts fail to provide the instance-specific adaptivity necessary for handling non-stationary field dynamics.

We propose **LASER** (*Learning Active Sensing for Continuum Field Reconstruction*), a unified, closed-loop framework that tightly integrates sparse field recovery, latent dynamics modeling, and active sensing (as shown in Table 1). We formalize active sensing as a Partially Observable Markov Decision Process (POMDP), where sensor locations are treated as controllable actions. The core of our solution to this POMDP is a **continuum field latent world model**.

[1]MoE Key Lab of Artificial Intelligence, Institute of AI, School of Computer Science, Shanghai Jiao Tong University. Correspondence to: Yunbo Wang <yunbow@sjtu.edu.cn>.

*Proceedings of the $43^{rd}$ International Conference on Machine Learning*, Seoul, South Korea. PMLR 306, 2026. Copyright 2026 by the author(s).

*Table 1.* **Comparison of continuum field reconstruction and sensing methods.** Unlike existing approaches, LASER uniquely integrates forward prediction with active, optimized sensor placement to handle sparse observations in a closed-loop framework.

| Model | Sparse Obs. | Forward Predict | Placement Optim. | Active Sensing |
|---|---|---|---|---|
| DiffusionPDE (2024) | ✗ | ✗ | ✗ | ✗ |
| AROMA (2024) | ✓ | ✓ | ✗ | ✗ |
| PhySense (2025) | ✓ | ✗ | ✓ | ✗ |
| **LASER (Ours)** | ✓ | ✓ | ✓ | ✓ |

We refer to this architecture as a "world model" because it serves as a surrogate of the physical environment. It not only captures the underlying spatiotemporal dynamics but also provides intrinsic reward feedback based on reconstruction fidelity. This world model enables the reinforcement learning (RL) agent to simulate "what-if" sensing scenarios and plan optimal movements by predicting the field's future evolution within a latent space.

More concretely, as illustrated in Figure 1, the RL policy is trained to generate proactive sensor displacements at each time step by conditioning on both the current sparse measurements and the predicted latent states provided by the world model. Upon action execution, the agent queries new observations from the training dataset, while rewards are computed based on the reconstruction fidelity evaluated by the world model. This mechanism forms a closed "sensing-reconstruction-evaluation" optimization loop without requiring interaction with a live physical system.

The primary contributions of this work are as follows:

- We introduce LASER, an RL-based framework that formalizes sensor placement as a POMDP. Our method enables instance-specific sensing strategies in response to evolving physical conditions.

- We develop a latent "world model" for continuum fields that integrates field reconstruction with latent forward modeling. This predictive architecture allows the sensing policy to "look ahead" and navigate toward high-information regions beyond current observations.

- Extensive experiments demonstrate that LASER achieves superior fidelity across diverse physical domains. It is shown to outperform existing approaches with fixed or offline-optimized sensing layouts consistently.

## 2. Related Work

Learning-based methods have shown strong modeling capability for continuum fields (Long et al., 2018; Kochkov et al., 2021; Pfaff et al., 2021; Li et al., 2020; 2021; Han et al., 2022; Serrano et al., 2024). These methods span a diverse set of architectural paradigms, including CNN models tailored to dense and uniform grids (Stachenfeld et al., 2022; Ummenhofer et al., 2020; Guan et al., 2022;

Zhu et al., 2024), GNNs that naturally support irregular meshes and flexible discretizations (Battaglia et al., 2016; Pfaff et al., 2021; Janny et al., 2023; Deng et al., 2025), implicit neural representations that enable continuous space-time modeling (Yin et al., 2023; Luo et al., 2024; Serrano et al., 2023), and transformer-based operators that support efficient global reasoning and fast solution of parametric PDEs (Hao et al., 2023; Wu et al., 2022; 2024; Alkin et al., 2024; Iakovlev et al., 2023; Steeven et al., 2024). Collectively, these models achieve high predictive accuracy and strong generalization, but typically rely on idealized assumptions of dense and complete observations, limiting their applicability in realistic settings with sparse, irregular, noisy, or partial measurements.

Recent models begin to address field reconstruction under sparse observations, but assume fixed, predefined sensing locations across time and trajectories (Serrano et al., 2024; Luo et al., 2024; Steeven et al., 2024; Iakovlev & Lähdesmäki, 2025). DiffusionPDE (Huang et al., 2024) conditions score-based generative models (Song et al., 2021) on sparse observations, but relies on random or fixed layouts and incurs a high computational cost due to the need for thousands of sampling steps. PhySense (Ma et al., 2025) optimizes sensor locations offline over a global training distribution, yielding static layouts that lack instance-level adaptivity to non-stationary dynamics (Wu et al., 2022; Deng et al., 2025). Moreover, both lines of work focus on per-step reconstruction and do not support autoregressive rollout with dynamically adapting sensing locations.

## 3. Problem Definition

We consider a dynamical physical system characterized by a continuum field $\boldsymbol{u}_t : \Omega \to \mathbb{R}^C$ defined over a spatial domain $\Omega \subset \mathbb{R}^d$, where $C$ represents the number of physical quantities (*e.g.*, velocity, pressure). The system evolves with unknown, non-stationary spatiotemporal dynamics: $\boldsymbol{u}_{t+1} = \mathcal{F}(\boldsymbol{u}_t)$. In practice, the full field $\boldsymbol{u}_t$ is not directly observable. Instead, we obtain sparse, pointwise measurements at $N$ **controllable** sensor locations $\boldsymbol{X}_t = \{\boldsymbol{x}_t^{(i)}\}_{i=1}^N$:

$$\boldsymbol{o}_t = \left\{ \left( \boldsymbol{x}_t^{(i)}, \boldsymbol{u}_t(\boldsymbol{x}_t^{(i)}) \right) \right\}_{i=1}^N, \ \boldsymbol{x}_t^{(i)} \in \Omega, \ N \ll |\mathcal{D}|, \quad (1)$$

where $\mathcal{D}$ is a discrete candidate set of possible sensor positions. Our objective is to learn an active sensing policy $\pi$ that sequentially optimizes sensor displacements $\boldsymbol{a}_t = \Delta \boldsymbol{X}_t$ to minimize the long-term reconstruction error across $\Omega$:

$$\min_{\pi,\phi} \ \mathbb{E}_{\boldsymbol{u}_{1:T}} \left[ \sum_{t=1}^T \mathcal{L} \left( \boldsymbol{u}_t, \ \hat{\boldsymbol{u}}_t(\boldsymbol{o}_{1:t}^\pi; \ \phi) \right) \right], \quad (2)$$

where $\hat{\boldsymbol{u}}_t$ is the field reconstructed by a parametrized model $\phi$ from the history of observations gathered under the policy $\pi$ up to time $t$, and $\mathcal{L}$ denotes the mean squared error (MSE) evaluated over the full discretized spatial domain $\mathcal{D}$: $\frac{1}{|\mathcal{D}|} \sum_{\boldsymbol{x} \in \mathcal{D}} \|\boldsymbol{u}_t(\boldsymbol{x}) - \hat{\boldsymbol{u}}_t(\boldsymbol{x})\|_2^2$.

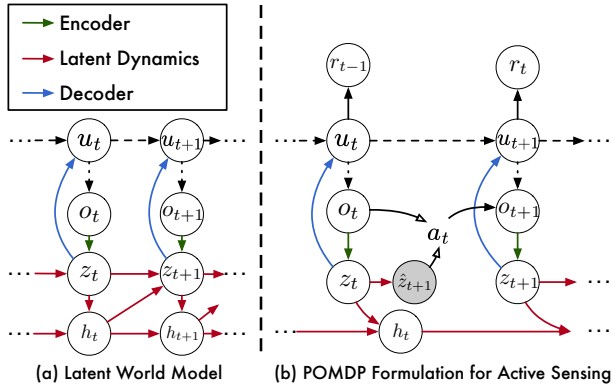

*Figure 2.* **The LASER graphical model.** (a) Latent world model with jointly trained encoder, dynamics, and decoder. (b) Active sensing as a POMDP, where the policy interacts with the world model and receives updated latent states and rewards.

## 4. Method

In this section, we introduce the technical details of our LASER method. Unlike traditional passive reconstruction pipelines, LASER treats sensor placement as an online, adaptive process governed by an RL policy. At its core, the framework leverages a continuum field latent world model to capture the underlying spatiotemporal dynamics of the field $\boldsymbol{u}_t$. We formalize this process as a POMDP defined over the latent states of the world model, where the agent's actions are informed by "imagined" future states.

Sec. 4.1 formalizes active sensing over continuum fields as a POMDP and motivates the transition to a latent-based decision framework. Sec. 4.2 describes the architecture of the continuum field latent world model, which provides the predictive transitions and intrinsic rewards. Sec. 4.3 and Sec. 4.4 present the policy network architecture and the policy optimization algorithm.

### 4.1. Active Sensing as a Latent POMDP

As illustrated in Figure 2, we define the POMDP by the tuple $\mathcal{M} = (\mathcal{S}, \mathcal{A}, \mathcal{O}, \mathcal{E}, \mathcal{T}_\phi, \mathcal{R}_\phi, \gamma)$ as follows:

- Latent state $\boldsymbol{s}_t = [\boldsymbol{z}_t, \boldsymbol{h}_t]$: The agent models the latent state at time $t$ as the combination of the current latent code $\boldsymbol{z}_t$, which encodes the most recent sparse observations, and a recurrent history $\boldsymbol{h}_t$, which maintains a memory of the field's evolution.

- Action $\boldsymbol{a}_t$: The sensing action corresponds to the displacement vector $\Delta \boldsymbol{X}_t$ for the sensor array, which is continuous and bounded by practical sensor movement constraints.

- Observation $\boldsymbol{o}_t$ and the emission function $\mathcal{E}$: As defined in Eq. (1), we obtain $\boldsymbol{u}_t(\boldsymbol{x}_t^{(i)})$ at continuous coordinates $\boldsymbol{x}_t^{(i)}$ via bilinear interpolation from the nearest grid points, as $\boldsymbol{u}_t$ is defined on a discrete grid in our experiments.

- Latent transition $\mathcal{T}_\phi$: The world model predicts the evolution of the latent state: $\boldsymbol{z}_{t+1} \sim p_\phi^{\mathrm{dyn}}(\cdot \mid \boldsymbol{z}_t, \boldsymbol{h}_{<t})$.

- Reward $r_t$: The reward signal is defined as the negative reconstruction error evaluated by the world model's decoder: $r_t = -\mathcal{L}\big(\boldsymbol{u}_{t+1}, \hat{\boldsymbol{u}}_{t+1}(\boldsymbol{o}_{t+1}^\pi; \phi)\big)$.

- Discount factor $\gamma \in [0, 1)$: The factor balances immediate and long-term future rewards.

**Proactive sensing via latent imagination.** Effective sensing requires the agent to anticipate future field states rather than merely reacting to past measurements. Consequently, as shown in Figure 2(b), we condition the policy $\pi_\theta$ on the predicted future latent state $\hat{\boldsymbol{z}}_{t+1}$, effectively enabling the agent to "imagine" the impending field configuration:

$$\boldsymbol{a}_t \sim \pi_\theta(\cdot \mid \hat{\boldsymbol{z}}_{t+1}, \boldsymbol{o}_t), \quad \hat{\boldsymbol{z}}_{t+1} \sim p_\phi^{\mathrm{dyn}}(\cdot \mid \boldsymbol{z}_t, \boldsymbol{h}_{<t}). \quad (3)$$

This design shifts the sensing paradigm from *reactive* (sampling the current field) to *proactive* (anticipating and targeting informative regions of the future field), enabling the agent to strategically position sensors where they will be most valuable for the impending reconstruction task.

### 4.2. Continuum Field Latent World Model

We develop a continuum field reconstruction model $\phi$ that serves as a high-fidelity surrogate of the physical environment. Following the paradigm established in Model-Based RL, notably the "World Models" of Ha & Schmidhuber (2018), our architecture is characterized as a world model by its integration of three essential components:

1. Latent representation module ($p_\phi^{\mathrm{enc}}$): Our encoder maps sparse, irregular observations into compressed latent states to capture the underlying physics of global fields.

2. State transition module ($p_\phi^{\mathrm{dyn}}$): We implement a latent dynamics module that predicts the next-step latent state $\hat{z}_{t+1}$ conditioned on current history.

3. Reward evaluation module: Our Gaussian decoder ($p_\phi^{\mathrm{dec}}$) reconstructs the full continuum field from the latent state, directly computing the reconstruction fidelity that serves as a differentiable reward signal.

**Sparse observation encoder.** Like Serrano et al. (2024), our encoder maps a variable set of sparse observations $\boldsymbol{o}_t = \{(\boldsymbol{x}_t^{(i)}, \boldsymbol{u}_t(\boldsymbol{x}_t^{(i)}))\}$ to a fixed size Gaussian latent variable $\boldsymbol{z}_t \in \mathbb{R}^{M \times d_z}$. A set of learnable latent queries $\mathbf{Q} \in \mathbb{R}^{M \times d_q}$ attends to embeddings of the observation coordinates and values, yielding a permutation-invariant representation. This enables the encoder to process an arbitrary number of sensors at arbitrary locations, thereby forming the agent's "belief" state: $\boldsymbol{z}_t \sim p_\phi^{\mathrm{enc}}(\cdot \mid \boldsymbol{o}_t) = \mathcal{N}\big(\boldsymbol{\mu}_\phi(\boldsymbol{o}_t), \boldsymbol{\sigma}_\phi^2(\boldsymbol{o}_t)\big)$.

**Latent dynamics predictor.** To enable proactive decision-making (see Eq. (3)), the agent must reason about the *future* state of the field. We introduce a GRU module to maintain a recurrent history $\boldsymbol{h}_t = \mathrm{GRU}_\phi(\boldsymbol{h}_{t-1}, \boldsymbol{z}_t)$, where $\boldsymbol{h}_t$ is zero-initialized at $t = 0$. The dynamics predictor is then implemented as a conditional diffusion model (Peebles &

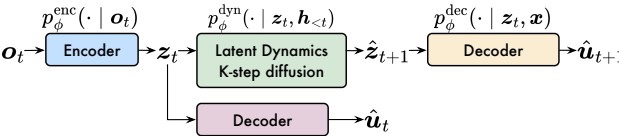

*Figure 3.* **Data flow of LASER world model.** Sparse observations $\boldsymbol{o}_t$ are encoded into a latent state $\boldsymbol{z}_t$, propagated forward to $\hat{\boldsymbol{z}}_{t+1}$ via the dynamics module, and finally decoded at spatial coordinates $\boldsymbol{x}$ to reconstruct the continuous field $\hat{\boldsymbol{u}}$.

Xie, 2023) that forecasts the next latent state by conditioning on both the current $\boldsymbol{z}_t$ and the historical summary $\boldsymbol{h}_t$: $\hat{\boldsymbol{z}}_{t+1} \sim p_\phi^{\mathrm{dyn}}(\cdot \mid \boldsymbol{z}_t, \boldsymbol{h}_t)$. Concretely, the model denoises a noise-corrupted latent $\tilde{\boldsymbol{z}}_{t+1}$ over $K$ steps. The resulting predictive latent $\hat{\boldsymbol{z}}_{t+1}$ incorporates trend information beyond the immediate observation, providing the policy with a forward-looking basis for deciding where to sense next.

**Continuum field decoder and reward evaluation.** The probabilistic Gaussian decoder mirrors the design in AROMA and ENMA (Koupaï et al., 2025): It is an implicit neural representation (an MLP) that conditions on $\boldsymbol{z}_t$ and any spatial coordinate $\boldsymbol{x} \in \Omega$ to reconstruct the field continuously: $\hat{\boldsymbol{u}}_t(\boldsymbol{x}) = p_\phi^{\mathrm{dec}}(\cdot \mid \boldsymbol{z}_t, \boldsymbol{x})$. By evaluating the reconstruction error against high-fidelity ground truth, the decoder defines the reward $r_t = -\mathcal{L}(\boldsymbol{u}_{t+1}, \hat{\boldsymbol{u}}_{t+1})$, aligning policy optimization with full-field accuracy.

**Training.** Figure 3 shows the module-level data-flow of the latent world model. The world model is trained entirely offline on a dataset of high-fidelity spatiotemporal field sequences $\boldsymbol{u}_{1:T}$. Unlike AROMA, where sensor locations are fixed per trajectory, we randomly sample a new set of sensor locations $\boldsymbol{x}_t^{(i)}$ uniformly from $\Omega$. This forces the model to learn a representation that is invariant to specific sensor layouts, thereby generalizing better to the diverse observation patterns encountered during active sensing. The model is trained by maximizing a variational lower bound, combining a reconstruction loss, a latent regularization term, and the diffusion loss for the dynamics:

$$\mathcal{L}_{\mathrm{world}}(\phi) = \mathbb{E}\Big[ \underbrace{- \log p_\phi^{\mathrm{dec}}(\boldsymbol{u}_t \mid \boldsymbol{z}_t, \boldsymbol{x})}_{\mathcal{L}_{\mathrm{recon}}}$$
$$+ \beta \cdot \underbrace{\mathcal{D}_{\mathrm{KL}}\big(p_\phi^{\mathrm{enc}}(\boldsymbol{z}_t \mid \boldsymbol{o}_t) \parallel \mathcal{N}(0, I)\big)}_{\text{KL divergence}}$$
$$+ \lambda \cdot \underbrace{\|\boldsymbol{\epsilon} - \boldsymbol{\epsilon}_\phi(\tilde{\boldsymbol{z}}_{t+1}, k, \boldsymbol{z}_t, \boldsymbol{h}_t)\|^2}_{\mathcal{L}_{\mathrm{diffusion}}} \Big], \quad (4)$$

where the negative log-likelihood term reduces to an MSE reconstruction objective, $\boldsymbol{\epsilon}$ is the true noise added in the diffusion forward process, and $\boldsymbol{\epsilon}_\phi$ is the denoising network.

### 4.3. Policy Network

Based on the pre-trained latent world model $\phi$, the policy network receives the predictive latent state $\hat{\boldsymbol{z}}_{t+1}$ from the

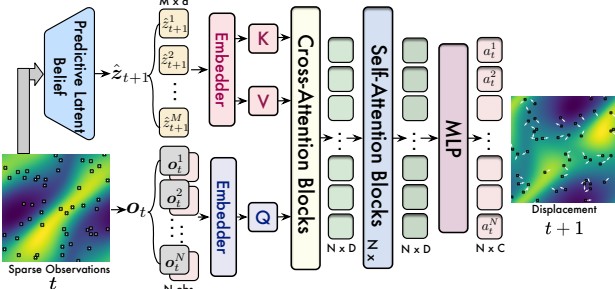

*Figure 4.* **The LASER policy architecture.** The network employs a cross-attention mechanism to fuse the predicted latent state $\hat{\boldsymbol{z}}_{t+1}$ and current sparse observations $\boldsymbol{o}_t$, outputting continuous sensor displacements for the next time step.

world model's dynamics module and the current sparse observations $\boldsymbol{o}_t$, and outputs continuous sensor displacement actions $\boldsymbol{a}_t = \Delta \boldsymbol{X}_t$. As shown in Figure 4, $\pi_\theta$ is implemented as a Transformer-based architecture designed to handle variable sensor configurations. The policy is parameterized as a conditional Gaussian distribution, with mean $\boldsymbol{\mu}_\theta$ and log-standard deviation $\log \boldsymbol{\sigma}_\theta$ output by the network. To capture fine-grained spatial dependencies, we encode sensor coordinates using a multi-scale Fourier feature embedding:

$$\gamma_{\mathrm{pos}}(\boldsymbol{x}) = \Big[\boldsymbol{x}, \{\gamma^s(\boldsymbol{x})\}_{s \in \mathcal{S}}\Big], \quad (5)$$
$$\gamma_{\mathrm{pos}}^s(\boldsymbol{x}) = \Big[\sin(\boldsymbol{x}\omega^s), \cos(\boldsymbol{x}\omega^s)\Big], \quad (6)$$

where $\omega^s$ are log-uniformly sampled frequencies at scale $s$. Each query vector $\mathbf{q}^{(i)} = [\gamma_{\mathrm{pos}}(\boldsymbol{x}_t^{(i)}); \mathrm{Embed}(\boldsymbol{u}_t(\boldsymbol{x}_t^{(i)}))]$ is formed by concatenating positional encodings and embedded sparse observations, producing $\mathbf{Q} \in \mathbb{R}^{N \times d_q}$.

The core reasoning occurs through a *multi-scale cross-attention mechanism*, where the encoded queries interact with the predictive latent state $\hat{\boldsymbol{z}}_{t+1}$. Here, the world model's "imagined" future serves as the key and value context:

$$\mathbf{q}^{(s)} = \mathrm{linear}_q^{(s)}(\mathbf{q}), \quad \mathbf{k}^{(s)}, \mathbf{v}^{(s)} = \mathrm{linear}_{k,v}^{(s)}(\hat{\boldsymbol{z}}_{t+1}), \quad (7)$$
$$\mathbf{f} = \bigoplus_{s \in \mathcal{S}} \mathrm{softmax}\left(\frac{\mathbf{q}^{(s)}(\mathbf{k}^{(s)})^\top}{\sqrt{c_s}}\right) \mathbf{v}^{(s)}, \quad (8)$$

where $\oplus$ denotes concatenation across scales. This interaction allows the policy to weigh current measurements against the anticipated global dynamics.

Following cross-attention, a series of self-attention blocks capture inter-sensor interactions to ensure coordinated movement across the array, yielding the feature set $\mathbf{f}'$. Finally, an MLP head projects these features to produce sensor displacement parameters $\boldsymbol{\mu}_\theta, \log \boldsymbol{\sigma}_\theta = \mathrm{Proj}(\mathbf{f}') \in \mathbb{R}^{N \times 2d}$, where $d$ is the spatial dimension of displacements. Actions are sampled as $\boldsymbol{a}_t^{(i)} \sim \mathcal{N}\big(\boldsymbol{\mu}_\theta^{(i)}, \mathrm{diag}((\boldsymbol{\sigma}_\theta^{(i)})^2)\big)$, and clipped to $[-a_{\max}, a_{\max}]$ to respect practical movement constraints. This architecture enables LASER to strategically reposition

**Algorithm 1** The training algorithm of LASER.

---

Initialize policy parameters $\theta$, old policy $\theta_{\text{old}} \leftarrow \theta$.
Initialize dynamic filtering threshold $\tau \leftarrow -\inf$.
**while** *not converged* **do**
  Initialize $\mathcal{D} \leftarrow \emptyset$.
  **for** *episode* $e = 1..E$ **do**
    Sample initial field $\boldsymbol{u}_{t_0}$ and sensor locations $\boldsymbol{x}_{t_0}$.
    **for** *time step* $t = t_0 \rightarrow t_0 + T - 1$ **do**
      Encode observation and predict next latent $\hat{\boldsymbol{z}}_{t+1}$.
      `// Dynamic group filtering with`
      `   look-ahead rewards`
      **for** *group* $g = 1..G$ **do**
        Sample action $\boldsymbol{a}_t^g \sim \pi_{\boldsymbol{\theta}}(\cdot \mid \hat{\boldsymbol{z}}_{t+1}, \boldsymbol{o}_t)$.
        Perform forward rollout $\{\hat{\boldsymbol{z}}_{t+h}\}_{h=1}^H$.
        Compute lookahead $r_t^{\text{lookahead},g}$ with Eq. (12).
      Compute minima $R_{\min}$.
      **if** $\forall g, r_t^{\text{lookahead},g} < \tau$ *and iteration*$> 1$ **then**
        **Exclude this group.**
      Store $(\hat{\boldsymbol{z}}_{t+1}, \boldsymbol{o}_t, \{\boldsymbol{a}_t^g, r_t^g\}_{g=1}^G)$ in $\mathcal{D}$.
      Update history $\boldsymbol{h}_{t+1} \leftarrow \text{GRU}(\boldsymbol{h}_t, \boldsymbol{z}_t)$.
  Update quality threshold $\tau \leftarrow \text{mean}(\{R_{\min}\})$.
  `// Policy optimization`
  **for** *epoch* $e = 1..N_{epoch}$ **do**
    Sample mini-batch $\mathcal{B} \sim \mathcal{D}$.
    Compute GRPO objective $\mathcal{J}_{\text{GRPO}}(\theta)$ with Eq. (11).
    Update policy parameters $\theta \leftarrow \theta - \eta \nabla_\theta \mathcal{J}_{\text{GRPO}}(\theta)$.
  Update old policy $\theta_{\text{old}} \leftarrow \theta$.

---

sensors by jointly reasoning about anticipated field dynamics, current measurements, and spatial context.

## 4.4. Reinforcement Learning Algorithm

In LASER, the RL agent learns to maximize reconstruction fidelity by interacting with the latent world model. LASER leverages an on-policy policy gradient method to solve the POMDP defined in Sec. 4.1. The overall learning procedure is briefly summarized in Algorithm 1.

**Environment interaction.** To prevent overfitting to specific initial conditions, each training episode begins by randomly selecting a trajectory and a starting timestep $t_0$ from the offline dataset. The environment is reset to the ground-truth field $\boldsymbol{u}_{t_0}$, and sensors are initialized with a uniform spatial distribution over $\Omega$. Throughout the episode, the world model's recurrent state $\boldsymbol{h}_t$ is updated with each new encoded $\boldsymbol{z}_t$ to reflect the evolving dynamics.

**Policy optimization with GRPO.** We employ Group Relative Policy Optimization (GRPO) (Guo et al., 2025; Yu et al., 2025; Zheng et al., 2025; Liu et al., 2026) to optimize $\pi_\theta$. GRPO is an on-policy RL algorithm that inherits the stability of Proximal Policy Optimization (PPO) (Schulman et al., 2017) while replacing the traditional value function with group-relative advantage estimation. At each timestep $t$, we sample $G$ independent action groups $\{\boldsymbol{a}_t^g\}_{g=1}^G$ from the current policy $\pi_\theta(\cdot|\hat{\boldsymbol{z}}_{t+1}, \boldsymbol{o}_t)$. Each action in the group

leads to a transition to the next state and yields an immediate reward $r_t^g$. The advantage $A_{g,t}$ is calculated by normalizing $r_t^g$ against the statistics of the entire group:

$$A_{g,t} = \frac{r_t^g - \text{mean}(\{r_t^{g'}\}_{g'=1}^G)}{\text{std}(\{r_t^{g'}\}_{g'=1}^G)}, \qquad (9)$$

where the mean and standard deviation are computed across the $G$ sampled rewards at that timestep. To maintain stable gradient estimates, we apply a secondary normalization step within each training mini-batch $\mathcal{B}$ (Liu et al., 2026; Zhu et al., 2026) where $\mu_{\mathcal{B}} = \text{mean}\{A_{g,t} \mid (A_{g,t}) \in \mathcal{B}\}$ and $\sigma_{\mathcal{B}} = \text{std}\{A_{g,t} \mid (A_{g,t}) \in \mathcal{B}\}$:

$$\hat{A}_{g,t} = \frac{A_{g,t} - \mu_{\mathcal{B}}}{\sigma_{\mathcal{B}} + \epsilon_{\text{norm}}}. \qquad (10)$$

Here, $\epsilon_{\text{norm}}$ is a small constant for numerical stability. The policy network is then updated by maximizing:

$$\mathcal{J}_{\text{GRPO}}(\theta) = \mathbb{E}_{(\hat{\boldsymbol{z}}_{t+1}, \boldsymbol{o}_t, \boldsymbol{a}_t^g, r_t^g) \sim \mathcal{D}_{\theta_{\text{old}}}} \left[ \frac{1}{G} \sum_{g=1}^G \frac{1}{N} \sum_{i=1}^N \right.$$
$$\left. \min\left(s_{g,t}(\theta)\hat{A}_{g,t}, \text{clip}(\cdot, 1-\epsilon, 1+\epsilon)\hat{A}_{g,t}\right) \right], \quad (11)$$

where $s_{g,t}(\theta) = \frac{\pi_\theta(\boldsymbol{a}_t^{i,g}|\hat{\boldsymbol{z}}_{t+1}, \boldsymbol{o}_t)}{\pi_{\theta_{\text{old}}}(\boldsymbol{a}_t^{i,g}|\hat{\boldsymbol{z}}_{t+1}, \boldsymbol{o}_t)}$ is the importance sampling ratio, $\theta_{\text{old}}$ denotes the policy parameters before the update, and $\epsilon$ is a clipping hyperparameter. This group-relative approach directly contrasts the information gain of different movement strategies.

**Dynamic group filtering.** Standard GRPO treats all sampled groups equally. However, in active sensing, certain action configurations may be systematically uninformative, such as sensors clustering in low-variance regions, leading to noisy gradient estimates. To this end, we propose dynamic group filtering. For each group, we calculate the minimum reward over group $R_{\min} = \min_g\{r_t^g\}$. Groups that consistently fall below a running mean of $R_{\min}$ are identified as low-quality and excluded from the rollout buffer. This mechanism allows the optimization process to remain focused on high-quality, informative trajectories.

**Predictive reward rollouts.** To transcend one-step GRPO optimization, LASER evaluates the impact of an action over a future horizon. After executing $\boldsymbol{a}_t$ to obtain the proposed sensor placement $\boldsymbol{X}_{t+1}$, we freeze this configuration and use the world model's latent dynamics module $p_\phi^{\text{dyn}}$ to perform an $H$-step rollout conditioned on the current history, where $H = 3$ in our experiments. This yields a sequence of predicted latent states $\{\hat{\boldsymbol{z}}_{t+k}\}_{h=1}^H$. Each predicted state is decoded to $\hat{\boldsymbol{u}}_{t+k}$ to evaluate its reconstruction fidelity, forming a weighted lookahead reward:

$$r_t^{\text{lookahead}} = \frac{\sum_{h=1}^H \gamma^{h-1} r_{t+h}}{\sum_{h=1}^H \gamma^{h-1}}. \qquad (12)$$

This reward formulation directly encourages LASER to place sensors in locations that will remain informative over multiple future time steps.

# 5. Experiments

We evaluate LASER across three datasets with distinct physical patterns, including *(i)* Turbulent Navier-Stokes Flow, *(ii)* Shallow-Water Equations, and *(iii)* real-world global Sea Surface Temperature (SST) with irregular land constraints.

*Table 2.* **Summary of benchmarks.**

| Benchmark | Geometry | #Dim | Resolution |
|-----------|----------|------|------------|
| Navier-Stokes | Regular Grid | 2D | $64 \times 64$ |
| Shallow-Water | Regular Grid | 3D | $64 \times 128$ |
| SST | Land Constraints | 3D | $171 \times 360$ |

## 5.1. Experimental Setup

As presented in Table 2, our experiments encompass both datasets in the 2D and 3D domains: ● **2D Navier-Stokes Equation** (Li et al., 2021; Constantin & Foiaş, 1988): The equation is expressed with the vorticity form on the unit torus, $\frac{\partial w}{\partial t} + u \cdot \nabla w = \nu \Delta w + f, \nabla u = 0$ for $x \in \Omega, t > 0$, where $\nu$ is the viscosity coefficient. We consider two different versions $\nu = 10^{-3}$ ($NS_{\nu 1e-3}$) and $\nu = 10^{-5}$ ($NS_{\nu 1e-5}$). ● **3D Shallow-Water Equation** (Serrano et al., 2023; Yin et al., 2023): This equation approximates fluid flow on the Earth's surface. The data includes the vorticity $w$ and height $h$ of the fluid. ● **Sea Surface Temperature** (Jean-Michel et al., 2021; Ma et al., 2025): daily sea surface temperature at approximately $1°$ resolution from the GLORYS12 dataset with intricate land-sea constraints.

We evaluate LASER from two perspectives to assess both its forward modeling fidelity and its decision-making efficacy: ● **Rollout assessment for the latent world model ($\phi$):** We first evaluate the reconstruction and prediction performance of the world model under randomly sampled sensor configurations. In this scenario, we compare LASER only with methods that support forward prediction. ● **Online reconstruction with active sensing ($\theta$):** We then assess the coupled performance of the field reconstruction and the adaptive sensor placement policy. We compare LASER against the state of the art from Table 1, including AROMA (Serrano et al., 2024), DiffusionPDE (Huang et al., 2024), and PhySense (Ma et al., 2025). Furthermore, we include an RL baseline, termed LASER$_{\text{ppo}}$, in which the proposed sensing strategy is replaced with a standard PPO policy (Schulman et al., 2017). See more details in Appendix D and E.

## 5.2. Experiments on Simulated PDE Datasets

**Setups.** We evaluate our method on the Navier-Stokes and Shallow-Water benchmarks under a temporal extrapolation setting. Each trajectory is divided into two equal-length segments: the first half, denoted as *In-t*, and the second half, denoted as *Out-t*. The model is trained exclusively on the *In-t* segment, using a budget of 256 randomly sampled sensor locations. At test time, the model autoregressively rolls

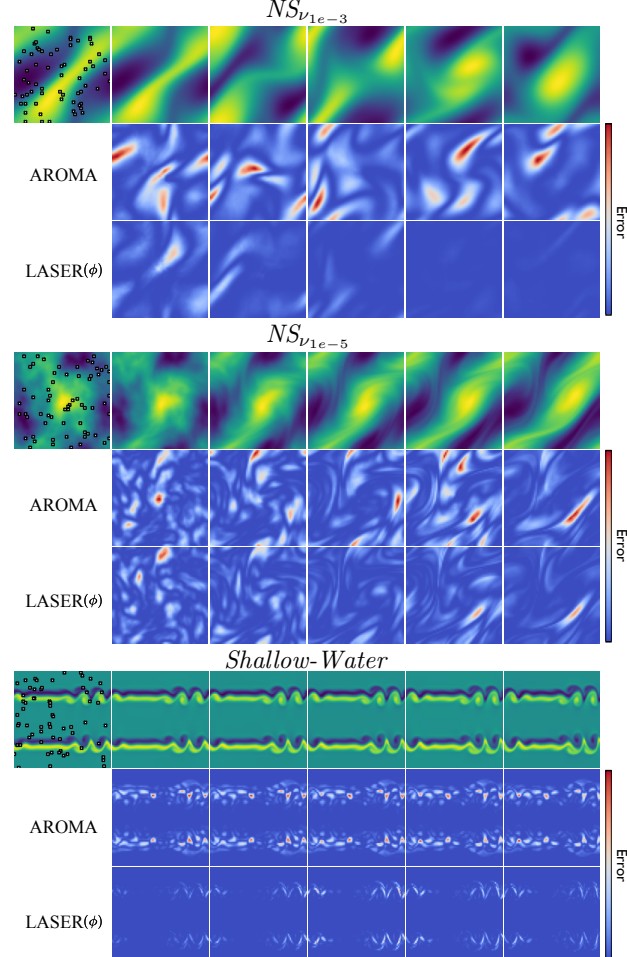

*Figure 5.* **Long-term rollout performance under high sparsity** ($N = 64$). We highlight the predictive errors of the LASER latent world model during extended temporal horizons.

out the learned dynamics starting from $t = 0$ over the entire *Out-t* horizon. Performance within the *In-t* interval reflects the model's forecasting accuracy under the training regime, while performance on *Out-t* measures its ability to extrapolate beyond the training horizon. To further stress-test the framework's robustness, we evaluate prediction performance under varying sensing budgets ($N \in \{256, 128, 64\}$). This assesses the generalization ability when subjected to reduced sensing coverage and unseen sensor layouts.

**Dynamics rollout performance.** We train the latent world model, LASER($\phi$), with randomly sampled sensor locations over the spatial domain at every training step. At test time, we randomly sample an initial sensor configuration at $t = 0$ and use the same placement. Given the resulting $o_{t=0}$, we encode it into the latent space and perform autoregressive latent rollouts using the learned dynamics model $p_\phi^{\text{dyn}}$ without receiving any further observations. We evaluate the rollout error by comparing the predicted fields against the ground-truth fields: $\text{MSE}_{\text{rollout}} = \frac{1}{T} \sum_{t=1}^{T} \frac{1}{|\mathcal{D}|} \sum_{x \in \mathcal{D}} \|\hat{u}_t(x) - u_t(x)\|_2^2$. As shown in **Ta-**

*Table 3.* **Rollout error** ($MSE_{rollout}$) **on the *Navier-Stokes* and *Shallow-water* benchmarks.** The same randomly sampled initial sensor placement at $t = 0$ is used for all methods, and rollouts are conditioned on sparse observations with $o_{t=0}$

| # Obs. ↓ | dataset → | $NS_{\nu 1e-3}$ ($\times 10^{-1}$) | | | $NS_{\nu 1e-5}$ ($\times 10^{-1}$) | | | *Shallow-Water* ($\times 10^{-2}$) | | |
|---|---|---|---|---|---|---|---|---|---|---|
| | Model ↓ | *In-t* | *Out-t* | *Avg* | *In-t* | *Out-t* | *Avg* | *In-t* | *Out-t* | *Avg* |
| 256 | AROMA (2024) | 0.191 | 0.635 | 0.413 | 0.180 | 5.876 | 3.028 | 1.472 | 1.739 | 1.606 |
| | LASER ($\phi$) | **0.006** | **0.090** | **0.050** | **0.073** | **5.228** | **2.730** | **0.006** | **0.074** | **0.050** |
| 128 | AROMA (2024) | 0.279 | 0.860 | 0.570 | 0.202 | **5.932** | 3.067 | 1.227 | 1.369 | 1.298 |
| | LASER ($\phi$) | **0.024** | **0.290** | **0.157** | **0.108** | 6.007 | **3.057** | **0.016** | **0.113** | **0.064** |
| 64 | AROMA (2024) | 0.683 | 1.519 | 1.101 | 0.267 | 6.166 | 3.217 | 1.293 | 1.476 | 1.385 |
| | LASER ($\phi$) | **0.209** | **0.589** | **0.399** | **0.196** | **6.081** | **3.138** | **0.138** | **0.274** | **0.206** |

*Table 4.* **Reconstruction error** ($MSE_{recon}$) **under static and active sensing strategies.** AROMA, DiffusionPDE, and LASER($\phi$) are evaluated using fixed sensor layouts. The final three models (PhySense, $LASER_{ppo}$, and LASER) utilize optimized sensing trajectories, in which $LASER_{ppo}$ replaces the sensing policy optimization method with PPO.

| # Obs. ↓ | dataset → | $NS_{\nu 1e-3}$ ($\times 10^{-3}$) | | | $NS_{\nu 1e-5}$ ($\times 10^{-1}$) | | | *Shallow-Water* ($\times 10^{-3}$) | | |
|---|---|---|---|---|---|---|---|---|---|---|
| | Model ↓ | *In-t* | *Out-t* | *Avg* | *In-t* | *Out-t* | *Avg* | *In-t* | *Out-t* | *Avg* |
| 256 | AROMA (2024) | 2.180 | 3.260 | 2.720 | 0.065 | 1.909 | 0.987 | 12.64 | 12.55 | 12.59 |
| | DiffusionPDE (2024) | 1.339 | 1.349 | 1.344 | 0.887 | 1.302 | 1.094 | 2.734 | 3.617 | 3.175 |
| | LASER ($\phi$) | 0.258 | 0.764 | 0.478 | 0.019 | 1.198 | 0.608 | 0.053 | 0.833 | 0.443 |
| | PhySense (2025) | 0.114 | 0.611 | 0.376 | 0.018 | 1.202 | 0.610 | 0.050 | 0.661 | 0.355 |
| | $LASER_{PPO}$ | 0.103 | 0.510 | 0.304 | 0.019 | **1.162** | **0.590** | 0.046 | 0.607 | 0.326 |
| | LASER | **0.096** | **0.483** | **0.302** | **0.017** | 1.188 | 0.603 | **0.042** | **0.465** | **0.257** |
| 128 | AROMA (2024) | 5.971 | 5.661 | 5.816 | 0.125 | 2.443 | 1.284 | 14.78 | 14.47 | 14.63 |
| | DiffusionPDE (2024) | 7.651 | 5.567 | 6.609 | 1.257 | 1.724 | 1.490 | 4.977 | 7.097 | 6.037 |
| | LASER ($\phi$) | 0.562 | 1.070 | 0.818 | 0.033 | 1.382 | 0.707 | 0.079 | 0.764 | 0.421 |
| | PhySense (2025) | 0.180 | 0.622 | 0.370 | **0.027** | 1.360 | 0.693 | 0.058 | 0.685 | 0.369 |
| | $LASER_{PPO}$ | **0.134** | 0.561 | 0.353 | 0.032 | 1.346 | 0.688 | 0.064 | 0.627 | 0.345 |
| | LASER | 0.149 | **0.493** | **0.321** | 0.031 | **1.343** | **0.688** | **0.052** | **0.492** | **0.299** |
| 64 | AROMA (2024) | 21.90 | 18.65 | 20.27 | 0.328 | 3.596 | 1.962 | 15.63 | 17.27 | 16.45 |
| | DiffusionPDE (2024) | 5.924 | 7.162 | 6.543 | 0.921 | 2.369 | 1.645 | 6.921 | 9.354 | 8.138 |
| | LASER ($\phi$) | 0.229 | 0.821 | 0.524 | 0.058 | 1.717 | 0.887 | 0.344 | 1.500 | 0.920 |
| | PhySense (2025) | 0.196 | 0.736 | 0.466 | 0.055 | 1.681 | 0.868 | **0.057** | 0.766 | 0.411 |
| | $LASER_{PPO}$ | 0.193 | **0.599** | **0.396** | 0.056 | **1.527** | **0.790** | 0.083 | 0.829 | 0.456 |
| | LASER | **0.169** | 0.726 | 0.434 | **0.055** | 1.559 | 0.801 | 0.133 | **0.587** | **0.386** |

ble 3, LASER($\phi$) consistently outperforms AROMA across all datasets and sensing budgets, with more pronounced gains under more sparsity. **Figure 5** presents rollout predictions based solely on the initial sparse observations on 64 sensing locations. LASER($\phi$) yields lower prediction errors. We provide more results in Appendix F.3.

**Active sensing results.** We use a pretrained continuum world model with 256 sparse observations and train separate policies or optimization-based models on top of it. We fix the sensor configuration sampled at $t = 0$ and collect sparse observations for $t > 0$ for AROMA, DiffusionPDE, and LASER($\phi$). For PhySense, following its original setup, we use an offline-optimized and instance-invariant sensor placement for $t > 0$. We evaluate the reconstruction error $MSE_{recon}$, where $p_\phi$ reconstructs the full field conditioned on the current sparse observations acquired at every time step. **Table 4** reports field reconstruction error with different sensing strategies. We observe that: First, active sensing approaches consistently outperform those with fixed sensor

layouts, with the performance gap widening as the number of observations decreases, highlighting the importance of adaptive sensing under sparse measurements. Second, LASER achieves the lowest reconstruction error by using a latent world model to anticipate future dynamics and actively adapt sensing trajectories at inference, ensuring robust generalization under sparse and shifting observations. **Figure 6** presents qualitative results with 64 sensing locations. More visualizations are shown in Appendix F.5.

### 5.3. Experiments on Real-World SST Dataset

The SST dataset contains temperature records sampled at random months and days across multiple years. Both the month and day are provided as additional input features. We perform full-sequence forecasting, predicting the entire temperature trajectory using 100 sampled sensor locations and evaluate performance under varying sensing budgets ($N \in 100, 50$). **Table 5** provides both *(i)* field prediction results from initial sparse observations and *(ii)* online recon-

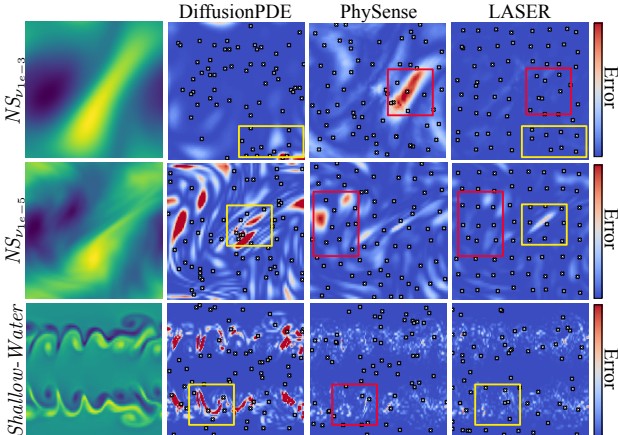

*Figure 6.* **Qualitative showcases of active sensing.** We evaluate different placement strategies under extreme sparsity ($N = 64$).

*Table 5.* **MSE results on the SST dataset under varying sensing budgets.** We report both *(i)* field prediction from initial sparse observations and *(ii)* online reconstruction with active sensing.

| Setting | Model | # Obs. | |
|---|---|---|---|
| | | 100 | 50 |
| $\mathrm{MSE_{rollout}}$ | AROMA (2024) | 6.3734 | 6.8064 |
| | LASER ($\phi$) | **2.6792** | **2.7310** |
| $\mathrm{MSE_{recon}}$ | AROMA (2024) | 1.0586 | 1.1281 |
| | LASER ($\phi$) | 0.7277 | 0.7402 |
| | DiffusionPDE (2024) | 3.4626 | 5.7087 |
| | PhySense (2025) | 0.7059 | 0.7287 |
| | LASER | **0.6932** | **0.7184** |

struction results with active sensing. LASER consistently outperforms compared methods on both tasks. **Appendix Figure 18** shows the qualitative results.

### 5.4. Model Analyses

**Ablation study on length of historical inputs.** We study the effect of the input history length used by the GRU-based latent dynamics predictor with 256 sensors. As shown in **Table 6**, incorporating historical latent features consistently improves rollout prediction compared to using no history, especially for out-of-distribution timesteps, highlighting the importance of history-aware latent dynamics modeling. We observe that the optimal history length depends on the turbulence level of the scene. For more turbulent regimes (see Figure 5), such as $NS_{\nu 1e-5}$ and *Shallow-water*, shorter history lengths achieve lower errors. This suggests that excessive historical context may introduce outdated or noisy information in highly chaotic systems, making shorter temporal contexts more effective for accurate rollout prediction.

**Ablation study on dynamic quality filtering.** We evaluate the effect of dynamic quality filtering on the final performance of LASER. We compare the full LASER model with two variants: *(i)* LASER$^\dagger$, a model without dynamic

*Table 6.* **Ablation study on temporal history for the latent dynamics predictor.** We evaluate $\mathrm{MSE_{rollout}}$ across varying GRU history lengths ($h_t$) within $p_\phi^{\mathrm{dyn}}$ using 256 sensors.

| Dataset → | $NS_{\nu 1e-3}$ ($\times 10^{-2}$) | | $NS_{\nu 1e-5}$ ($\times 10^{-1}$) | | *Shallow-Water* ($\times 10^{-3}$) | |
|---|---|---|---|---|---|---|
| History len. | *In-t* | *Out-t* | *In-t* | *Out-t* | *In-t* | *Out-t* |
| 0 | 0.088 | 0.822 | 0.078 | 5.38 | 0.074 | 1.104 |
| 3 | 0.087 | 0.904 | **0.073** | **5.23** | **0.064** | **0.774** |
| 4 | **0.063** | **0.904** | 0.079 | 6.80 | 0.078 | 0.889 |
| 5 | 0.069 | 1.012 | 0.097 | 7.88 | 0.069 | 0.893 |

*Table 7.* **Ablation study on dynamic quality filtering.** LASER$^\dagger$ denotes a baseline model without dynamic quality filtering. Results are reported in $\mathrm{MSE_{recon}}$ ($\times 10^{-3}$) on $NS_{\nu 1e-3}$.

| # Obs. | Model | *In-t* | *Out-t* | *Avg* |
|---|---|---|---|---|
| 256 | LASER ($\phi$) | 0.1907 | 0.5274 | 0.3591 |
| | LASER$^\dagger$ | 0.0973 | 0.6854 | 0.3913 |
| | LASER | **0.0958** | **0.4834** | **0.3024** |
| 128 | LASER ($\phi$) | 0.5617 | 1.0743 | 0.8180 |
| | LASER$^\dagger$ | 0.1361 | 0.8734 | 0.5048 |
| | LASER | **0.1490** | **0.4932** | **0.3211** |
| 64 | LASER ($\phi$) | 0.2285 | 0.8206 | 0.5245 |
| | LASER$^\dagger$ | **0.1499** | 0.7543 | 0.4521 |
| | LASER | 0.1693 | **0.7259** | **0.4339** |

quality filtering, and *(ii)* LASER($\phi$), a model without sensor displacement. As shown in **Table 7**, the final LASER model with dynamic quality filtering consistently improves overall performance, particularly in out-of-distribution timesteps. These results highlight the importance of dynamic quality filtering for robust active sensing, especially under sparse settings and highly non-stationary systems.

**Ablation study on the predictive reward.** We assess the effect of the forward-looking reward horizon $H$ on reconstruction accuracy in **Table 8**. We observe consistent improvements as the horizon increases from $H = 1$ to $H = 5$, indicating the benefit of evaluating sensing actions beyond immediate one-step outcomes. When $H=1$, the reward only reflects the reconstruction error at the next time step, leading to suboptimal sensor placements. In contrast, incorporating multi-step rollouts encourages the policy to account for the longer-term dynamics. This effect is particularly pronounced for *Out-t* predictions, where increasing $H$ from 1 to 5 yields a substantial reduction in error (from 0.6136 to 0.3380), suggesting that forward-looking rewards are crucial for generalization to future unseen states.

**Iterative co-training of the policy and the world model.** Active sensing performance relies heavily on the ability of the latent world model, which can degrade in highly chaotic regimes (*e.g.*, SST). To further push these limits, we evaluate an iterative co-training scheme: alternately fine-tuning the world model ($\phi$) on policy-collected observations and re-optimizing the policy ($\theta$) using updated predictive rewards. As shown in **Table 9**, this co-training consistently

*Table 8.* **Ablation study on the predictive reward.** We evaluate different forward-looking horizons for the reward function, and report the $\mathrm{MSE_{recon}}(\times 10^{-3})$ on $NS_{\nu 1e-3}$ with 256 sensors.

| Horizon | *In-t* | *Out-t* | *Avg* |
|---|---|---|---|
| 1 | 0.1104 | 0.6136 | 0.3620 |
| 3 | 0.0958 | 0.4834 | 0.3024 |
| 5 | **0.0938** | **0.3380** | **0.2813** |

*Table 9.* **Effect of iterative co-training.** $\mathrm{MSE_{recon}}$ on the SST dataset under varying sensing budgets to evaluate the joint adaptation of the world model and policy.

| Model | # Obs. | |
|---|---|---|
| | 100 | 50 |
| PhySense (Ma et al., 2025) | 0.7059 | 0.7287 |
| LASER | 0.6932 | 0.7184 |
| LASER (Co-train) | **0.6727** | **0.6934** |

reduces reconstruction errors. This demonstrates that jointly adapting the latent dynamics to the active sensing trajectories effectively mitigates model constraints in complex, non-stationary environments.

**Analysis of distributional shift.** Since the latent dynamics $p_\phi^{\mathrm{dyn}}$ is trained offline on randomly sampled sensors, a natural concern is whether it suffers from compounding errors under out-of-distribution, policy-induced trajectories during inference. To quantify this effect, we evaluate the world model's rollout performance on the $NS_{\nu 1e-3}$ dataset ($N$=256). Specifically, we compare autoregressive rollouts initialized at $t = 1$ using sensor locations induced by the trained policy (conditioned on initial random observations at $t = 0$) and perform autoregressive rollouts over the full trajectory against rollouts driven entirely by random sensors. As shown in **Table 10**, rather than degrading, the rollout performance actually improves under the policy-induced distribution, particularly in the extrapolated *Out-t* regime. This indicates the distribution shift is not harmful; rather, the policy identifies structurally critical locations that yield more accurate and stable latent predictions. Furthermore, iterative co-training yields additional gains, suggesting that coupling the world model updates with the policy-induced distribution can further eliminate potential shift penalties.

**Analysis of proactive sensing.** To investigate whether simpler sensor placement strategies can replace the proactive sensing strategy, we compare our sensing policy ($\theta$) with three heuristic alternatives, while leveraging the identical continuum world model ($\phi$) for field reconstruction:

- Entropy-based derives posterior entropy from the conditional diffusion world model, explicitly guiding sensors to regions where the model is least confident.

- Error-based acts as a reactive correction mechanism, targeting spatial locations that exhibited the highest reconstruction errors in the preceding time step.

*Table 10.* **Analysis of distributional shift.** $\mathrm{MSE_{rollout}}(\times 10^{-2})$ evaluated on the $NS_{\nu 1e-3}$ dataset with 256 sensors. We compare model rollouts driven by randomly sampled sensors versus policy-induced sensor placements.

| Sensor Distribution | *In-t* | *Out-t* | *Avg* |
|---|---|---|---|
| Random sampled | 0.061 | 0.971 | 0.525 |
| Policy-induced ($\theta$) | 0.058 | 0.498 | 0.284 |
| Iterative Co-training | **0.057** | **0.408** | **0.237** |

*Table 11.* **Proactive sensing vs. heuristic strategies.** $\mathrm{MSE_{recon}}$ ($\times 10^{-3}$) evaluated on $NS_{\nu 1e-3}$ under varying sensing budgets.

| Method | # Obs. | | |
|---|---|---|---|
| | 256 | 128 | 64 |
| Entropy-based | 1.387 | 7.467 | 52.50 |
| Error-based | 0.869 | 4.098 | 26.61 |
| Dynamics-based | 0.898 | 5.251 | 28.58 |
| LASER ($\theta$) | **0.096** | **0.149** | **0.169** |

- Dynamics-based prioritizes highly non-stationary regions by measuring the magnitude of temporal field changes (*i.e.*, $\|\hat{\boldsymbol{u}}_t - \hat{\boldsymbol{u}}_{t-1}\|_2$) to track rapid physical evolutions.

The next-step sensing locations are sampled proportionally to their respective spatial heuristic scores. As shown in **Table 11**, all three heuristics underperform, especially under extreme sparsity ($N = 64$), where errors increase by $10\times$–$100\times$ compared to LASER. These results highlight the limitation of heuristics: they blindly optimize instantaneous, reactive signals while ignoring the long-term spatiotemporal coupling of continuum fields. In contrast, LASER ($\theta$) proactively plans sensing trajectories based on latent imagination, making it essential. We provide visualizations in Figure 12 in Appendix F.4.

**Sensitivity analyses.** We evaluate LASER's sensitivity to initial sensor locations in Appendices F.1–F.2. The results demonstrate that once trained, LASER generates consistent outcomes based on various initial placements.

## 6. Conclusions and Limitations

We introduced LASER, a framework that learns active sensing for sparse continuum field reconstruction. By making decisions based on a predictive world model, LASER proactively navigates toward high-information regions beyond current observations. Experiments show that LASER outperforms prior sparse field reconstruction methods, enabling online adaptation in dynamic environments.

A potential limitation of LASER is its reliance on the latent world model's fidelity, where inaccurate dynamics can degrade sensing decisions in highly complex physical environments. Moreover, the optimization of sensing trajectories introduces extra computational overhead compared to static alternatives. We plan to resolve these issues in the future.

## Impact Statement

This work presents LASER, whose goal is to advance the field of machine learning by enabling more efficient active sensing for continuum field reconstruction in complex physical environments. By reducing the number of required observations while maintaining high reconstruction accuracy, LASER has the potential to support applications in scientific modeling, environmental monitoring, and autonomous systems operating under sensing constraints.

As a general-purpose learning framework, LASER does not introduce new ethical risks beyond those commonly associated with machine learning systems for perception and decision-making. Potential societal impacts will depend on downstream applications, and we do not foresee specific negative consequences arising directly from this work. We believe the contributions are primarily technical and methodological, with positive implications for data-efficient physical modeling.

## Acknowledgements

This work was supported by the National Natural Science Foundation of China (62250062), the Smart Grid National Science and Technology Major Project (2024ZD0801200), the Shanghai Municipal Science and Technology Major Project (2021SHZDZX0102), Shanghai Jiao Tong University AI for Engineering Initiative (WH410263001/005), and the Fundamental Research Funds for the Central Universities.

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

## A. Algorithm

We present the full algorithm of LASER in Alg. 2.

---

**Algorithm 2** The training algorithm of LASER.

---

Initialize policy parameters $\theta$, old policy $\theta_{\text{old}} \leftarrow \theta$.
Initialize dynamic filtering threshold $\tau \leftarrow -\inf$.
**while** *not converged* **do**

    // Rollout collection with quality filtering
    Initialize $\mathcal{D} \leftarrow \emptyset$.
    **for** *episode* $e = 1..E$ **do**
        Sample initial field $\boldsymbol{u}_{t_0}$, sensor locations $\boldsymbol{x}_{t_0}$, initialize history $\boldsymbol{h}_{t_0}$.
        **for** *time step* $t = t_0 \rightarrow t_0 + T - 1$ **do**
            Encode observation $\boldsymbol{z}_t \leftarrow p_\phi^{\text{enc}}(\boldsymbol{o}_t)$; predict next latent $\hat{\boldsymbol{z}}_{t+1} \leftarrow p_\phi^{\text{dyn}}(\boldsymbol{z}_t, \boldsymbol{h}_t)$.
            // Dynamic group sampling with quality filtering
            **for** *group* $g = 1..G$ **do**
                Sample action $\boldsymbol{a}_t^g \sim \pi_{\boldsymbol{\theta}}(\cdot \mid \hat{\boldsymbol{z}}_{t+1}, \boldsymbol{o}_t)$.
                // Forward-looking reward
                Execute $\boldsymbol{a}_t^g$, observe $\boldsymbol{o}_{t+1}^g$
                Forward rollout $\{\hat{\boldsymbol{z}}_{t+h}\}_{h=1}^H$; decode predictions and rewards $\{r_{t+h}^g\}$.
                Compute lookahead $r_t^{\text{lookahead},g}$ with Eq. 12.
            Compute group rewards $\{r_t^{\text{lookahead},g}\}_{g=1}^G$ and minima $R_{\min}$.
            **if** $\forall g, r_t^{\text{lookahead},g} < \tau$ **and** *iteration* $> 1$ **then**
                **continue**
            Store $(\hat{\boldsymbol{z}}_{t+1}, \boldsymbol{o}_t, \{\boldsymbol{a}_t^g, r_t^g\}_{g=1}^G)$ in $\mathcal{D}_{\text{new}}$.
            Update history $\boldsymbol{h}_{t+1} \leftarrow \text{GRU}(\boldsymbol{h}_t, \boldsymbol{z}_t)$.
    Update threshold $\tau \leftarrow \text{mean}(\{R_{\min}\})$.

    // Policy optimization
    **for** *epoch* $e = 1..N_{epoch}$ **do**
        Sample mini-batch $\mathcal{B} \sim \mathcal{D}$.
        Compute group-relative advantages via normalization within each group $A_{g,t}$ with Eq. 9.
        Normalize advantages across batch $\hat{A}_{g,t}$ with Eq. 10.
        Compute GRPO objective $\mathcal{J}_{\text{GRPO}}(\theta)$ with Eq. 11.
        Update policy parameters $\theta \leftarrow \theta - \eta \nabla_\theta \mathcal{J}_{\text{GRPO}}(\theta)$.
    Update old policy $\theta_{\text{old}} \leftarrow \theta$.

---

## B. Continuum Field Latent World Model

This section provides architectural and implementation details of the continuum field latent world model introduced in Sec. 4.2.

$$
\begin{aligned}
\text{Encoder:} && \boldsymbol{z}_t &\sim p_\phi^{\text{enc}}(\boldsymbol{z}_t \mid \boldsymbol{o}_t) \\
\text{Latnet Dynamics:} && \boldsymbol{z}_{t+1} &\sim p_\phi^{\text{dyn}}(\boldsymbol{z}_t \mid \boldsymbol{z}_t, \boldsymbol{h}_{<t}), \\
\text{Decoder:} && \boldsymbol{u}_t &\sim p_\phi^{\text{dec}}(\boldsymbol{z}_t).
\end{aligned}
\tag{13}
$$

### B.1. Sparse Observation Encoder

The encoder maps irregularly sampled observations of the physical field to a fixed-size latent token representation $\boldsymbol{z}_t \in \mathbb{R}^{M \times d_z}$ as $\boldsymbol{z}_t \sim p_\phi^{\text{enc}}(\cdot \mid \boldsymbol{o}_t) = \mathcal{N}(\boldsymbol{\mu}_\phi(\boldsymbol{o}_t), \boldsymbol{\sigma}_\phi^2(\boldsymbol{o}_t))$, where $M$ is a fixed number of latent tokens. This mapping is implemented using a sequence of cross-attention operations that aggregate information from the observed locations and

values, followed by a variational bottleneck. A set of learnable latent query tokens $\mathbf{Q} = (\mathbf{q}^1, \cdots, \mathbf{q}^M) \in \mathbb{R}^{M \times d_q}$ is introduced and shared across all training data. These tokens are initialized as trainable parameters and serve as queries in the cross-attention layers, enabling the encoder to compress a variable number of observations into a fixed-size latent representation.

**Encoding sparse obsevations** Each observation location $\boldsymbol{x}_t^i$ is embedded using a fourier feature embedding $\gamma(\cdot)$, and each observed value $\boldsymbol{u}_t(\boldsymbol{x}_t^i)$ is embedded using a learnable value embedding, yielding sequences of position embeddings $\gamma(\boldsymbol{x}_t^i)$ and value embeddings $\text{Embed}_u(\boldsymbol{u}_t(\boldsymbol{x}_t^i))$. $\gamma(\boldsymbol{x}_t^i)$ are defined as $\gamma_{\text{pos}}(\boldsymbol{x}_t^i) = \left[\boldsymbol{x}_t^i, \left\{\sin(\boldsymbol{x}_t^i \omega_l), \cos(\boldsymbol{x}_t^i \omega_l)\right\}_{l=1}^L\right]$ with $\omega_l = \pi \cdot b^{\ell_l}, \ell_l \in [0, L]$.

**Geometry encoding.** The geometry-aware representations $\mathbf{z}_t^{\text{geo}} \in \mathbb{R}^{M \times d_z}$ capture the spatial arrangement of the observation points at time $t$. They are obtained by applying a cross-attention mechanism between the set of learnable latent query tokens $\mathbf{Q} = (\mathbf{q}^1, \cdots, \mathbf{q}^M)$ to the positional embeddings of the observation coordinates via a cross-attention mechanism. The queries depend solely on the observation locations. This attention-based operation produces latent tokens that encode the geometric structure of the sampled points, such that similar spatial configurations yield similar geometry-aware representations, independent of the corresponding field values.

$$\mathbf{q}' = \text{Linear}_q(\mathbf{q}), \ \mathbf{k}, \mathbf{v} = \text{Linear}_{k,v}\left(\gamma(\boldsymbol{x}_t^i)\right), \ \mathbf{z}_t^{\text{geo}} = \mathbf{q} + \text{FFN}\left(\text{softmax}\left(\frac{\mathbf{q}'\mathbf{k}^\top}{\sqrt{c}}\right)\mathbf{v}\right). \tag{14}$$

**Observation encoding.** The observation-aware representations $\mathbf{z}_t^{\text{obs}} \in \mathbb{R}^{M \times d_z}$ integrate information from the observed field values, conditioned on the spatial geometry encoded in $\mathbf{z}_t^{\text{geo}}$ through a second cross-attention mechanism:

$$\mathbf{q}' = \text{Linear}_q(\mathbf{z}_t^{\text{geo}}), \ \mathbf{k}, \mathbf{v} = \text{Linear}_{k,v}\left(\gamma(\boldsymbol{x}_t^i), \text{Embed}_u(\boldsymbol{u}_t(\boldsymbol{x}_t^i))\right), \ \mathbf{z}_t^{\text{obs}} = \mathbf{z}_t^{\text{geo}} + \text{FFN}\left(\text{softmax}\left(\frac{\mathbf{q}'\mathbf{k}^\top}{\sqrt{c}}\right)\mathbf{v}\right), \tag{15}$$

where the keys encode the observation locations and the values encode the corresponding field measurements. This operation injects the physical field information into the latent tokens while preserving their geometry-aware structure.

**Variational bottleneck.** To obtain a compact and regularized latent representation, we introduce a variational bottleneck on the latent tokens. Concretely, the observation-aware representations $\mathbf{z}_t^{\text{obs}}$ are mapped to the parameters of a Gaussian distribution:

$$\boldsymbol{\mu}_t = \text{Linear}_\mu(\mathbf{z}_t^{\text{obs}}), \qquad \log \boldsymbol{\sigma}_t = \text{Linear}_\sigma(\mathbf{z}_t^{\text{obs}}), \tag{16}$$

from which the final latent tokens are sampled using the reparameterization trick $\boldsymbol{z}_t = \boldsymbol{\mu}_t + \boldsymbol{\sigma}_t \odot \boldsymbol{\epsilon}, \ \boldsymbol{\epsilon} \sim \mathcal{N}(\mathbf{0}, \mathbf{I})$. This stochastic bottleneck regularizes the latent space and encourages smooth latent representations, which is beneficial for learning stable long-term dynamics and improves robustness to sparse and noisy observations.

### B.2. Latent Dynamics Predictor

Our latent dynamics predictormodels the transition between consecutive latent states as a conditional diffusion process in the latent token space. The predictor takes the form: $\boldsymbol{z}_{t+1} \sim p_\phi^{\text{dyn}}(\cdot \mid \boldsymbol{z}_t, \boldsymbol{h}_t)$, where $\boldsymbol{z}_t \in \mathbb{R}^{M \times d}$ denotes the latent tokens at time $t$, and $\boldsymbol{h}_t$ is a recurrent hidden state summarizing historical information. We employ a GRU (Chung et al., 2014) to maintain a temporally coherent representation of past states: $\boldsymbol{h}_t = \text{GRU}_\phi(\boldsymbol{h}_{t-1}, \boldsymbol{z}_t)$, where $\boldsymbol{h}_t \in \mathbb{R}^{M \times d_h}$ is zero-initialized at $t = 0$. Our GRU processes each latent token independently while sharing parameters across tokens. This design allows each spatial token to maintain its own temporal context, capturing location-specific evolution patterns while enabling information flow across tokens through the shared GRU parameters. This hidden state captures temporal dependencies beyond the immediate observation, providing context for the diffusion process.

**Conditional diffusion.** We adopt a standard Gaussian diffusion formulation (Ho et al., 2020). The forward noising process gradually corrupts clean data $\boldsymbol{z}_{t+1}^0$ over $K$ steps: $q(\boldsymbol{z}_{t+1}^k \mid \boldsymbol{z}_{t+1}^0) = \mathcal{N}(\boldsymbol{z}_{t+1}^k;, \sqrt{\bar{\alpha}^k}, \boldsymbol{z}_{t+1}^0,, (1 - \bar{\alpha}^k)\mathbf{I})$, where $\{\bar{\alpha}^k\}_{k=1}^K$ follows a predefined noise schedule (Ho et al., 2020; Lippe et al., 2023; Peebles & Xie, 2023; Nichol & Dhariwal, 2021). We use an exponentially decreasing schedule as in Lippe et al. (2023). At diffusion step $k$, the model processes a concatenated latent sequence $\mathbf{x}_k = \left[\boldsymbol{z}_t; \tilde{\boldsymbol{z}}_{t+1}^k\right] \in \mathbb{R}^{2M \times d_z}$. The forward diffusion process is defined exclusively on the target latents $\tilde{\boldsymbol{z}}_{t+1}^k$, while the context latents $\boldsymbol{z}_t$ remain noise-free.

**Conditioning mechanism.** The diffusion transformer is conditioned on both the diffusion step $k$ and the GRU hidden state $h_t$. The diffusion step embedding $e_k = \text{Embed}(k) \in \mathbb{R}^{d_k}$ is concatenated with $h_t$ to form a unified conditioning vector: $c_k = [e_k; h_t] \in \mathbb{R}^{d_k + d_h}$.

This vector is projected through an MLP to produce the adaLN-Zero modulation parameters $(\gamma_k, \beta_k, g_k)$ that modulate all transformer blocks (Peebles & Xie, 2023).

**Noise prediction and training.** Given $x_k$ and $c_k$, the diffusion transformer predicts the added Gaussian noise $\epsilon_\theta = \epsilon_\theta(x_k, c_k) \in \mathbb{R}^{2M \times d_z}$. We denote by $\epsilon_\theta^k$ the prediction corresponding to the target tokens (the second half of the sequence). The model is trained using the simplified objective (Ho et al., 2020) $\mathcal{L}_{\text{diff}} = \mathbb{E}_{k, \epsilon} \left[ \| \epsilon_\theta^k(x_k, c_k) - \epsilon \|_2^2 \right]$, where $\epsilon \sim \mathcal{N}(0, \mathbf{I})$ and $k$ is uniformly sampled from $1, \ldots, K$.

**Reverse diffusion process.** During inference, we generate $\hat{z}_{t+1}$ by iteratively denoising from pure noise $\tilde{z}_{t+1}^K \sim \mathcal{N}(0, \mathbf{I})$. For $k = K, \ldots, 1$: $\tilde{z}_{t+1}^{k-1} = \frac{1}{\sqrt{\alpha_k}} \left( \tilde{z}_{t+1}^k - \sqrt{1 - \alpha_k}, \epsilon_\theta^k \right) + \sigma_k \xi$, where $\xi \sim \mathcal{N}(0, \mathbf{I})$. The final prediction is $\hat{z}_{t+1} = \tilde{z}_{t+1}^0$. This latent dynamics predictor enables the agent to reason about future field states while accounting for temporal dependencies, forming the basis for proactive sensing decisions.

### B.3. Continuum Field Decoder

The decoder implements an implicit neural representation that reconstructs the physical field continuously over the domain $\Omega$ from latent tokens and spatial coordinates. It takes the form: $\hat{u}_t(x) = p_\phi^{\text{dec}}(\cdot \mid z_t, x), \forall x \in \Omega$. This formulation enables querying the field value at any spatial location, providing a continuous reconstruction that serves both for field estimation and as the reward model in the RL framework. The decoder consists of three main components:

- Latent token processing. Given latent representation $z_t \in \mathbb{R}^{M \times d_z}$, we first apply a linear projection to increase the channel dimension, followed by self-attention blocks: $z_t' = \text{SelfAttn-Block}(\text{Linear}(z_t))$. This allows tokens to interact and exchange spatial information, producing enhanced representations $z_t'$.

- Multi-scale cross-attention. For a query coordinate $x \in \Omega$, we compute a multi-scale Fourier feature encoding $\gamma_{\text{pos}}(x)$ as defined in Eq. 6. Then, we compute a local feature vector $f_t(x)$ at the query location through multi-head cross-attention between the positional encoding and processed tokens with Eq. 8. The features from all scales are combined via concatenation and projection. This multi-scale design enables the decoder to capture both local details and global structures in the field.

- Implicit mapping. A lightweight multilayer perceptron maps the aggregated feature to the reconstructed field value $\hat{u}_t(x) = \text{MLP}(f_t(x))$. The MLP consists of three linear layers with a SiLU activation in between, producing predicted field measurement.

This decoder architecture follows the design principles of AROMA (Serrano et al., 2024) and ENMA (Koupaï et al., 2025), providing a continuous, coordinate-based representation that aligns the agent's objective with reconstruction fidelity across the entire spatial domain.

## C. Benchmarks

As presented in Table 2, our experiments encompass both datasets in the 2D and 3D domains. The benchmarks include:

- **2D Navier-Stokes Equation** (Li et al., 2021; Constantin & Foiaş, 1988): for a viscous and incompressible fluid. The equation is expressed with the vorticity form on the unit torus: $\frac{\partial w}{\partial t} + u \cdot \nabla w = \nu \Delta w + f$, $\nabla u = 0$ for $x \in \Omega, t > 0$, where $\nu$ is the viscosity coefficient. We consider two different versions $\nu = 10^{-3}$ ($NS_{\nu 1e-3}$) and $\nu = 10^{-5}$ ($NS_{\nu 1e-5}$). We have 256 trajectories of horizon 40 for training and 32 for testing for $NS_{\nu 1e-3}$ and 1000 trajectories of horizon 20 and 200 for testing for $NS_{\nu 1e-5}$.

- **3D Shallow-Water Equation** (Serrano et al., 2023; Yin et al., 2023): This equation approximates fluid flow on the Earth's surface. The data includes the vorticity $w$ and height $h$ of the fluid. The training set comprises 64 trajectories of size 40, and the test set comprises 8 trajectories with 40 timestamps.

- **Sea Temperature** (Jean-Michel et al., 2021): daily sea surface temperature at $1°$ resolution with intricate land constraints from the GLORYS12 reanalysis datasetincorporing intricate land–sea constraints. The training set consists of 216 trajectories, while the test set contains 49 trajectories with varying lengths of recorded observations, which are irregularly sampled in time.

## D. Compared Methods

We compare LASER with following methods:

- AROMA (Serrano et al., 2024) An encode-process-decode framework leveraging attention mechanisms, a latent diffusion transformer for spatio-temporal dynamics and neural fields for decoding, which can only place fixed sensors simultaneously.

- DiffusionPDE (Huang et al., 2024) A diffusion-based PDE solver that fill in missing information using generative priors to solve PDEs from partial observations, which requires full field input during model training. During inference, the sparse observations are predefined by random placement.

- PhySense (Ma et al., 2025) A synergistic two-stage framework integrating a flow-based generative reconstruction model with a sensor placement optimization strategy through projected gradient descent throughout the training distribution. PhySense lacks the ability of forward prediction and adapt sensing locations according to the input field.

- PPO (Schulman et al., 2017) A family of policy optimization methods in RL that use multiple epochs of stochastic gradient ascent to perform each policy update, proposing a novel objective with clipped probability ratios.

## E. Implementation Details

We conduct all experiments on a single RTX 3090 GPU. We introduce the implementation details of the continuum latent world model, policy optimization in Sec. E.1 and Sec. E.2. Furthermore, we detail the implementation of baselined in Sec. E.3.

### E.1. Training Details of Continuum Field Latent World Model

*Table 12.* Hyperparameters of the continuum latent world model for different datasets.

| Parameter | Notation | Dataset | | | |
|---|---|---|---|---|---|
| | | $NS_{\nu 1e-3}$ | $NS_{\nu 1e-5}$ | *Shallow-Water* | *Sea Surface Temperature* |
| `Sparse Observation Encoder` | | | | | |
| Number of latent queries | $M$ | 32 | 256 | 32 | 32 |
| Query embedding dimension | $d_q$ | 128 | 128 | 128 | 128 |
| Latent feature dimension | $d_z$ | 16 | 16 | 16 | 16 |
| Include geometry encoding | - | True | True | True | True |
| Include time encoding | - | - | - | - | True |
| Number of latent heads | $H_l$ | 4 | 4 | 4 | 4 |
| Latent head dimension | $d_l$ | 32 | 32 | 32 | 32 |
| KL-divergence loss weight | $\beta$ | $1e-5$ | $1e-5$ | $1e-5$ | $1e-5$ |
| `Latent Dynamics Predictor` | | | | | |
| Feature dimension | $d_f$ | 128 | 128 | 128 | 128 |
| Number of attention heads | - | 4 | 4 | 4 | 4 |
| Number of attention blocks | - | 4 | 4 | 4 | 4 |
| Modulation MLP dimension | - | 512 | 512 | 512 | 512 |
| History length | $T_h$ | 4 | 3 | 4 | 4 |
| Number of GRU layers | - | 2 | 2 | 2 | 2 |
| Diffusion loss weight | $\lambda$ | 1 | 1 | 1 | 1 |
| Diffusion steps | $K$ | 3 | 3 | 3 | 3 |
| `Continuum Field Decoder` | | | | | |
| MLP depth | - | 3 | 3 | 3 | 3 |
| MLP width | - | 128 | 128 | 64 | 128 |
| Number of self-attention blocks | - | 2 | 3 | 2 | 2 |
| Number of latent heads | $H_l$ | 4 | 4 | 4 | 4 |
| Latent head dimension | $d_l$ | 32 | 32 | 32 | 32 |
| Multi-scale frequency bands | $\mathcal{S}$ | $[2, 3]$ | $[3, 4, 5]$ | $[2, 3]$ | $[3, 4, 5]$ |
| Frequencies per scale | $N_f$ | 12 | 12 | 12 | 12 |
| Cross-attention feature dimension | $d_f$ | 16 | 16 | 16 | 16 |

We list the hyperparameters of the continuum field latent world model in Table 12. We perform a two-stage training: we first train the sparse observation encoder and continuum field decoder, then train the latent dynamics predictor, following AROMA's training strategies (Serrano et al., 2024).

In the first stage, the encoder-decoder is trained to reconstruct the field at each time step. The network learning rate is set to $1 \times 10^{-3}$, while the learnable latent queries $\mathbf{Q} \in \mathbb{R}^{M \times d_q}$ are optimized with a learning rate of $1 \times 10^{-2}$. Training uses a cosine annealing learning rate scheduler over 5000 epochs. In the second stage, the encoder and decoder parameters are frozen, and the latent dynamics predictor is trained on the encoded latent $\mathbf{z}$. The learning rate is set to $1 \times 10^{-3}$, and training proceeds with a cosine annealing scheduler for 2000 epochs.

During both stages, 256 observation points are randomly sampled at each iteration to obtain sparse observations. At test time, we evaluate rollout performance under different numbers of observations (256, 128, and 64) to assess the model's generalization capabilities across varying observation sparsity.

### E.2. Training Details of Active Sensing Policy

We list the hyperparameters for training the active sensing policy in Table 13. The policy outputs continuous actions. For the sensing spatial domain $\Omega \in (-1, 1)^2$, the maximum action magnitude is set to $a_{\max} = 0.05$. At each step, we manually clip out of bound sensor locations back to the spatial domain. At environment initialization, sparse observation locations are uniformly sampled over the spatial domain from a predefined set. During policy training, we adopt a linear-then-constant learning rate scheduler. The learning rate is linearly annealed from $1 \times 10^{-4}$ to $1 \times 10^{-5}$ over the first half of training and then kept constant for the remaining iterations. To prevent premature policy collapse and loss of exploration, the dynamic filtering mechanism is reset to $-\infty$ with probability $0.1$, such that all groups are retained in the subsequent rollout collection.

We additionally consider a PPO-based baseline (Schulman et al., 2017), in which the proposed sensing strategy is replaced by a standard PPO policy, referred to as LASER$_{\text{ppo}}$. For this baseline, the outputs of the final self-attention block are concatenated and fed into an additional value MLP to predict the state value $V(s_t)$. The learning rate of this value network is set to be ten times larger than that of the other policy modules. The PPO is updated by maximizing:

$$\mathcal{J}_{\text{PPO}}(\theta) = \mathbb{E}_t \Big[ \min \big( s_t(\theta) \hat{A}_t, \text{clip}(s_t(\theta), 1 - \epsilon, 1 + \epsilon) \hat{A}_t \big) - v_f \big( V_\theta(s_t) - \hat{V}_t \big)^2 \Big], \tag{17}$$

where $s_t(\theta)$ denotes the probability ratio between the current and old policies, and $\hat{A}_t$ is the estimated advantage. The $v_f$ is set to 5 in all experiments.

We use GRPO group size $G = 8$ and batch size 512 (64 groups). On-policy algorithms are optimized with $1 \times 10^6$ environment steps.

*Table 13.* Hyperparameters of active sensing policy architecture and training parameters.

| Name | Notation | Value |
|---|---|---|
| `Policy Architecture` | | |
| Number of self-attention blocks | $N_{\text{attn}}$ | 2 |
| Number of latent heads | - | 4 |
| Latent head dimension | - | 128 |
| Multi-scale frequency bands | $\mathcal{S}$ | $[2, 3]$ |
| Frequencies per scale | $N_f$ | 7 |
| `RL training parameters` | | |
| Training steps | - | $1 \times 10^6$ |
| Discount factor | $\gamma$ | 0.99 |
| Predictive rewards horizon | $H$ | 3 |
| Clip range | $\epsilon$ | 0.2 |
| Value loss weight (PPO) | $v_f$ | 5 |
| Maximum action | $a_{\max}$ | 0.05 |

### E.3. Baseline Details

**AROMA.** AROMA is trained using the same model hyperparameters as LASER ($\phi$). Following its original setup, the sensor locations are fixed for each trajectory throughout training. We randomly sample initial sensor placement during

inference.

**DiffusionPDE.** We borrow the original model architectures of DiffusionPDE, a SongUNet architecture (Song et al., 2021) with the EDMPrecond diffusion wrapper (Karras et al., 2022), as the network backbone for diffusion training without PDE constraints. DiffusionPDE incurs a substantial computational cost during inference, requiring tens to hundreds of seconds per frame, whereas our policy completes the process from location selection to frame reconstruction in approximately $0.5$ seconds. We reduce the number of diffusion steps to $500$ with negligible impact on performance.

**PhySense.** We adopt the same model architecture as LASER ($\phi$). PhySense is evaluated based on its reconstruction performance of the continuum field, where sensor placements are optimized over the training distribution. We optimize the sensor locations using projected gradient descent for $50$ epochs with a learning rate of $1$, since PhySense directly learns the sampling coordinates in the spatial domain. During inference, we use the optimized sensor placement for all test trajectories.

## F. Additional Results

### F.1. Sensitivity to Initial Sampled Locations

We analyze the sensitivity of LASER to the initialization of sampled sensor locations at the beginning of each test trajectory. After training the active sensing policy, we evaluate its robustness by varying the initial sensor placements at the first timestep while keeping the learned policy and continuum field reconstruction model fixed. Specifically, for each test sequence, we randomly initialize the sensor locations using different random seeds and then roll out the sensing policy to sequentially select future sensing locations and perform field reconstruction.

We report the mean and standard deviation of the prediction errors across different initializations in Table 14 over three random seeds. As shown in the table, LASER exhibits relatively low variance across different initial sensor placements under all sensing budgets and datasets, indicating that its performance is largely insensitive to the choice of initial sampled locations. While the reconstruction error increases as the number of observations decreases, the corresponding standard deviations remain small, suggesting that the learned sensing policy consistently guides sensors toward informative regions regardless of the initial configuration.

*Table 14.* **Sensitivity of LASER to initial sampled sensor locations.** We report the mean and standard deviation of prediction errors across three random initializations.

| # Obs. ↓ | $NS_{\nu 1e-3}$ ($\times 10^{-3}$) | | | $NS_{\nu 1e-5}$ ($\times 10^{-1}$) | | | Shallow-Water ($\times 10^{-3}$) | | |
| --- | --- | --- | --- | --- | --- | --- | --- | --- | --- |
| | *In-t* | *Out-t* | *Avg* | *In-t* | *Out-t* | *Avg* | *In-t* | *Out-t* | *Avg* |
| #256 | 0.096±3.79e-3 | 0.483±5.59e-2 | 0.302±7.05e-2 | 0.017±2.42e-3 | 1.188±1.16e-1 | 0.603±1.08e-1 | 0.042±4.08e-3 | 0.465±3.40e-2 | 0.257±2.58e-2 |
| #128 | 0.149±2.92e-3 | 0.493±3.23e-2 | 0.321±6.71e-2 | 0.031±1.54e-3 | 1.343±1.01e-1 | 0.688±1.22e-1 | 0.052±3.94e-3 | 0.492±4.01e-2 | 0.299±3.26e-2 |
| # 64 | 0.169±4.23e-3 | 0.726±4.18e-2 | 0.434±8.13e-2 | 0.055±1.82e-3 | 1.559±1.38e-1 | 0.801±1.62e-1 | 0.133±5.26e-3 | 0.587±3.99e-2 | 0.386±3.76e-2 |

### F.2. Sensitivity to Initial Placement Pattern

In our experiments, the sensor positions at time $t = 0$ are randomly initialized by uniformly sampling within the spatial domain and adding Gaussian noise $\mathcal{N}(0, 0.1)$. During evaluation, we investigate the impact of different initial sensor distributions on the final reconstruction performance. Specifically, we consider two settings: *(i)* an initialization scheme identical to that used during training, and *(ii)* a fully random initialization. Figure 7 visualizes the active sensing trajectories under these two different initialization conditions on $NS_{\nu 1e-3}$ with $256$ sparse observations.

From a quantitative perspective, the choice of initial sensor distribution has a limited effect on the final reconstruction error. With random initialization, the average MSE is $3.4449 \times 10^{-4}$, compared to $3.042 \times 10^{-4}$ under the training-consistent initialization. The slight performance degradation can be attributed to the fact that random sampling is more likely to produce duplicate or clustered sensor locations in the early stages, which reduces the effective spatial coverage of observations.

Furthermore, we zoom in on the sensor distributions at the final time step $t = 39$ (see Figure 8). We observe that, regardless of the initial sensor configuration, the policy gradually adjusts the sensing locations over time and converges to a similar spatial distribution in the final frame. This behavior indicates that the learned active sensing policy is robust to variations in initial conditions and can adaptively optimize sensor placement under different initial configurations, leading to stable reconstruction performance.

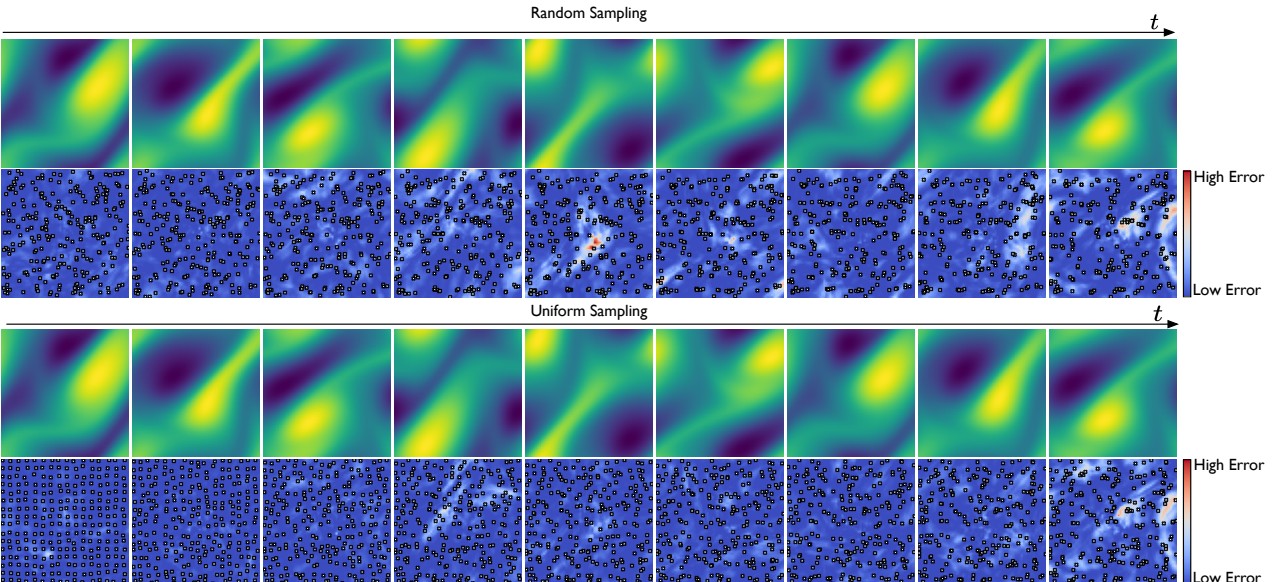

*Figure 7.* Showcases of different inital placement pattern on $NS_{\nu 1e-3}$ with 256 sparse observations.

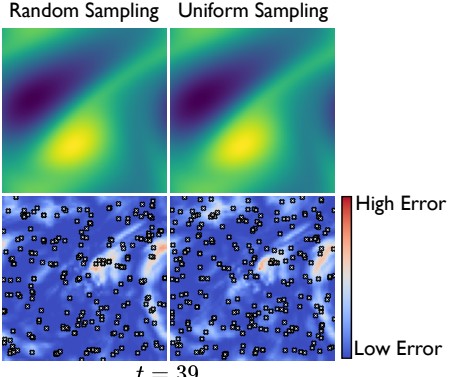

*Figure 8.* Comparison of final sensor distributions ($t = 39$) under different initial conditions

## F.3. Rollout Visualizations

We visualize the rollout performance comparing AROMA and LASER ($\phi$) in Figure 9-11. As shown in Table 3 and Figure 9-11, LASER ($\phi$) consistently outperforms AROMA across all benchmarks and under all levels of sparse observations. This is attributed to the various sensor location situations learned during training and the memory of the field's evolution maintained in LASER ($\phi$). Based solely on the initial sparse observations, LASER ($\phi$) yields low prediction errors throughout the sequence, both on *In-t* segments and *Out-t* segments, especially achieving significantly lower errors than AROMA in regions of the physical fields where values change rapidly. Although LASER ($\phi$) is only trained with a budget of 256 sensing locations, it still performs well under rollouts with 128 and 64 observations, demonstrating strong robustness and generalization capability.

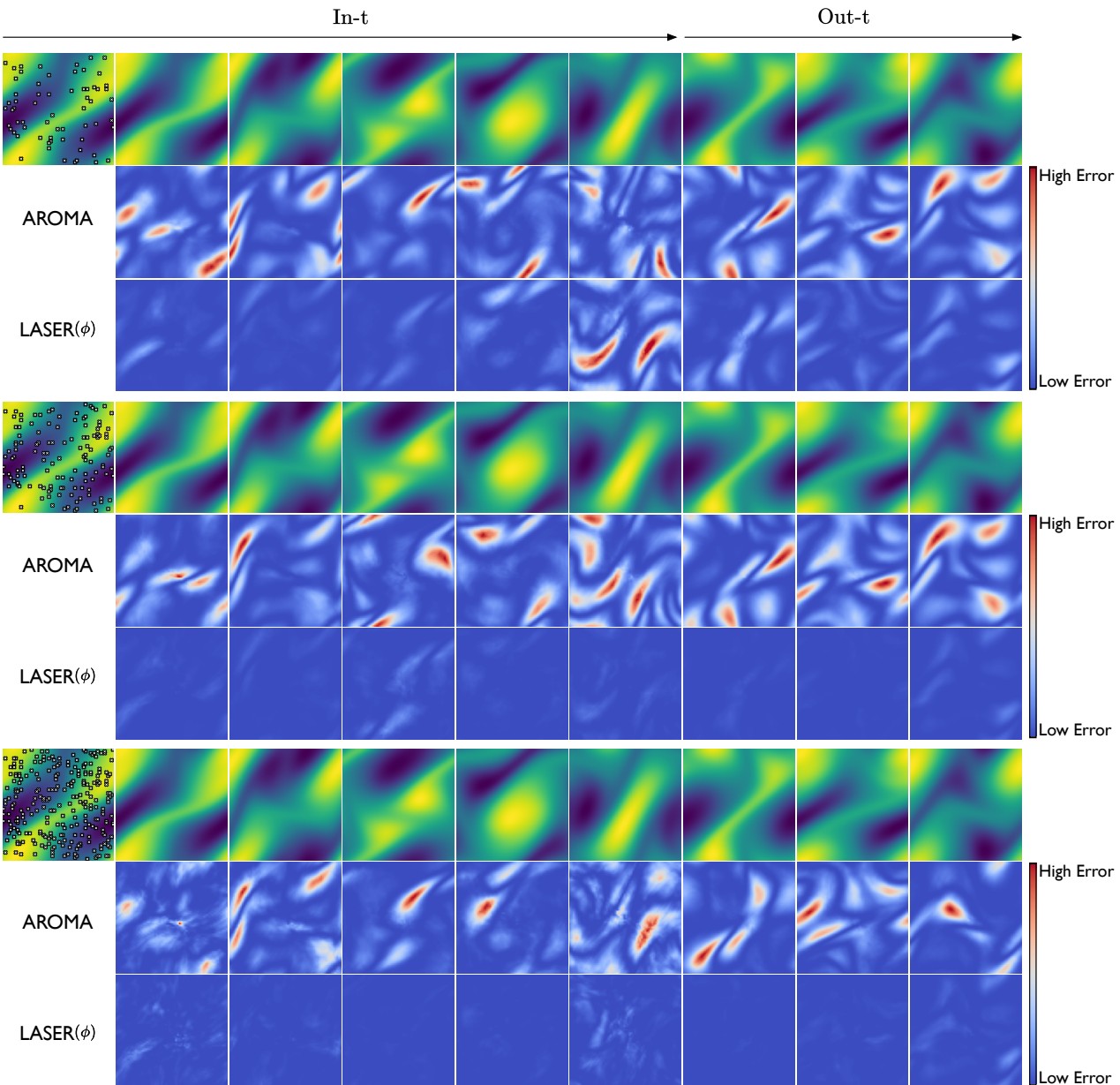

*Figure 9.* Showcases of rollout performance on $NS_{\nu 1e-3}$ across different sparse observations. Rows 1–3: 64 observations, Rows 4–6: 128 observations, Rows 7–9: 256 observations. The ground truth is presented in rows 1, 4 and 7, with the corresponding error maps shown directly below.

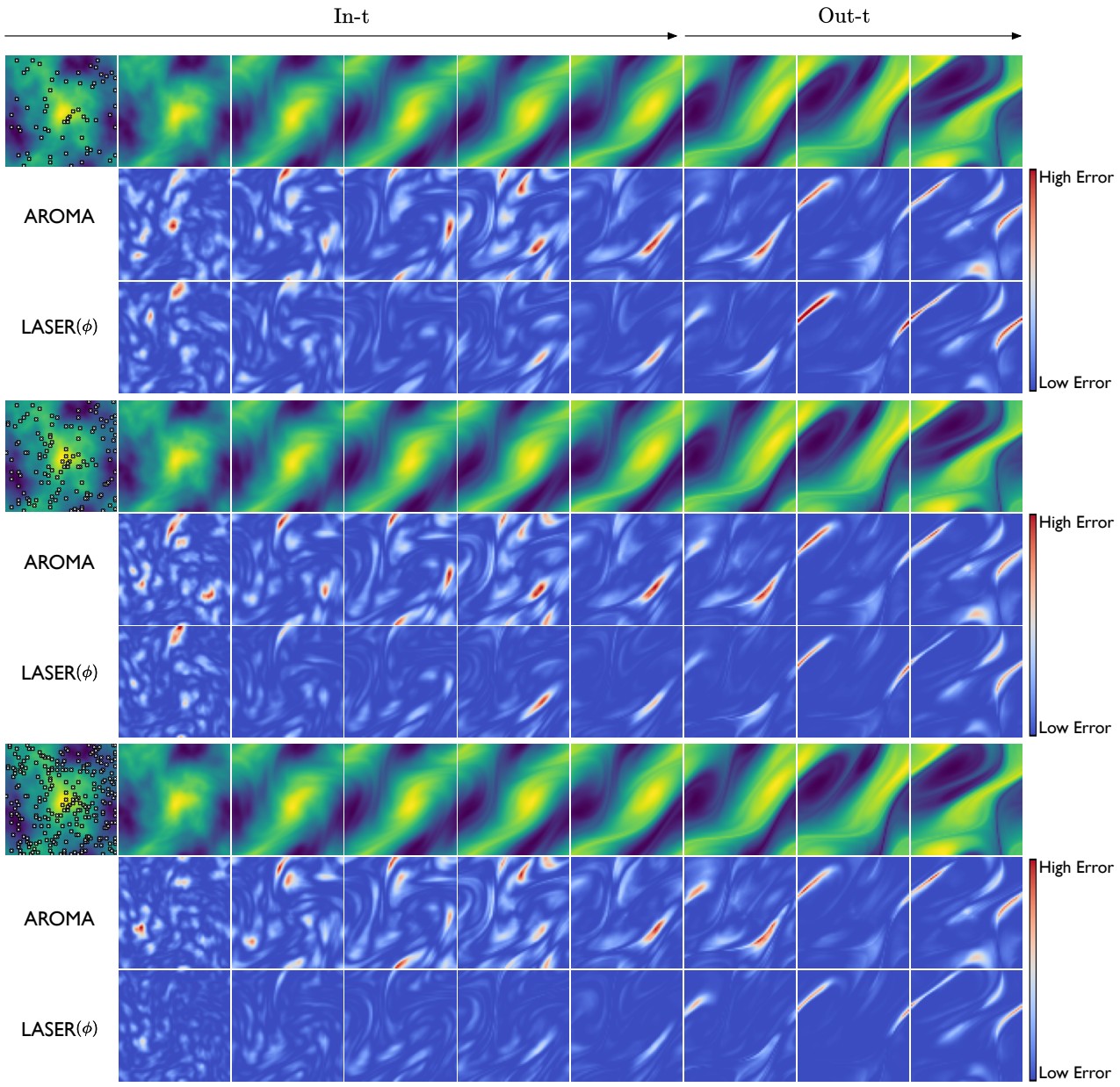

*Figure 10.* Showcases of rollout performance on $NS_{\nu 1e-5}$ across different sparse observations. Rows 1–3: 64 observations, Rows 4–6: 128 observations, Rows 7–9: 256 observations. The ground truth is presented in rows 1, 4 and 7, with the corresponding error maps shown directly below.

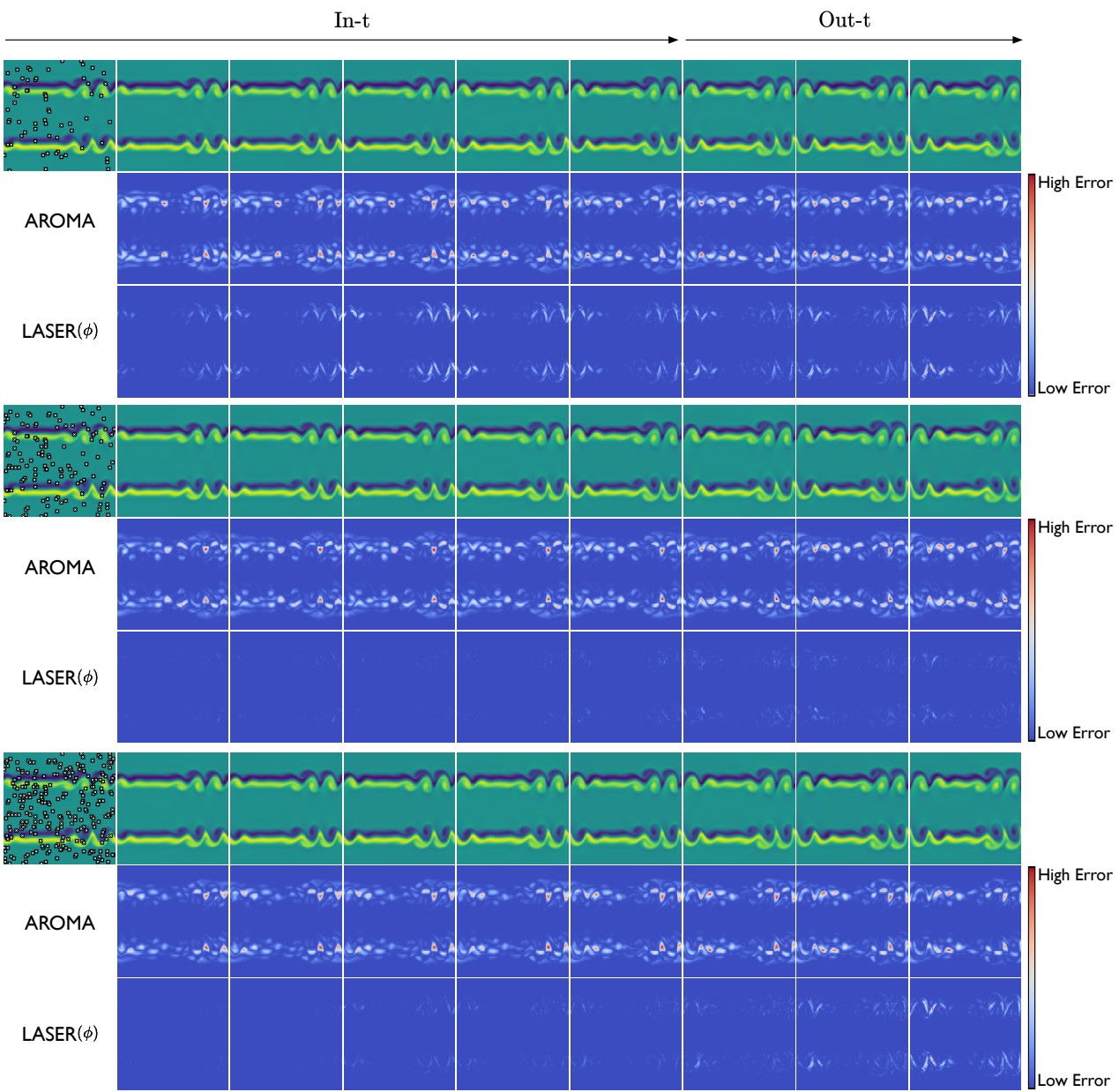

*Figure 11.* Showcases of rollout performance on *Shallow-Water* across different sparse observations. Rows 1–3: 64 observations, Rows 4–6: 128 observations, Rows 7–9: 256 observations. The ground truth is presented in rows 1, 4 and 7, with the corresponding error maps shown directly below.

### F.4. Comparison to Heuristic Sensing Strategies

We provide additional details for the heuristic sensing baselines. All heuristic baselines use the same trained continuum world model $\phi$ as LASER and differ only in how the next-step sensing locations are selected. Thus, the comparison isolates the effect of the sensing policy $\theta$.

At each sensing step, each heuristic produces a spatial score map $s_t(\boldsymbol{x})$ over candidate locations, and the next $N$ sensors are sampled proportionally to the normalized scores, $p_t(\boldsymbol{x}) = \frac{s_t(\boldsymbol{x})+\epsilon}{\sum_{\boldsymbol{x}'}(s_t(\boldsymbol{x}')+\epsilon)}$, where $\epsilon$ is a small constant for numerical stability. The entropy-based baseline uses the predictive uncertainty of the conditional diffusion world model as the score. The error-based baseline uses the previous-step reconstruction error, $s_t^{\mathrm{err}}(\boldsymbol{x}) = \|\hat{\boldsymbol{u}}_{t-1}(\boldsymbol{x}) - \boldsymbol{u}_{t-1}(\boldsymbol{x})\|_2$. The dynamics-based baseline uses the predicted temporal variation, $s_t^{\mathrm{dyn}}(\boldsymbol{x}) = \|\hat{\boldsymbol{u}}_t(\boldsymbol{x}) - \hat{\boldsymbol{u}}_{t-1}(\boldsymbol{x})\|_2$.

The quantitative comparison is reported in Table 11, where all heuristic strategies underperform LASER, especially under sparse sensing budgets. This indicates that directly optimizing instantaneous uncertainty, previous reconstruction error, or local temporal changes is insufficient for long-horizon spatiotemporal sensing. In contrast, LASER learns a proactive sensing policy through latent imagination, leading to more effective sensor placement.

Figure 12 further visualizes the sensing behavior of different strategies on the $NS_{\nu 1e-3}$ dataset with $N = 256$ sensors. The three columns show the ground-truth field, the selected sensing locations, and the corresponding reconstruction error map, respectively.

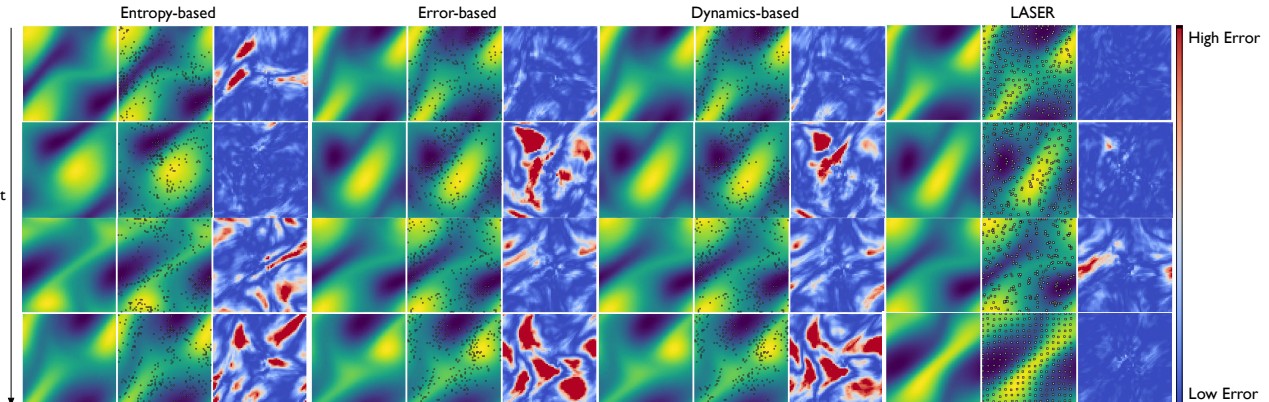

*Figure 12.* Qualitative comparison of LASER and heuristic sensing strategies on the $NS_{\nu 1e-3}$ dataset with $N = 256$ sensors. Compared to entropy-, error-, and dynamics-based heuristics, LASER places sensors more effectively for reconstructing the evolving continuum field.

### F.5. Active Sensing Visualizations

We present qualitative results with 64 sensing locations with different sensing strategies in Figure 13 on $NS_{\nu 1e-3}$, $NS_{\nu 1e-5}$ and *Shallow-Water* benchmarks. As shown in Table 4 and Figure 13, methods with fixed sensor layouts consistently underperform active sensing approaches, with the performance gap widening as the number of observations decreases, highlighting the importance of adaptive sensing under sparse measurements. DiffusionPDE relies on posterior sampling from sparse observations, and without PDE-based regularization, its reconstruction suffers especially at low observation budgets. PhySense optimizes sensors for the training distribution and fix them at inference, limiting adaptation and degrading performance. LASER achieves the lowest reconstruction error by using a latent world model to anticipate future dynamics and actively adapt sensing trajectories at inference, ensuring robust generalization under sparse and shifting observations.

The qualitative results with 100 sensing locations with different sensing strategies in Figure 14 on *Sea Surface Temperature* benchmarks. DiffusionPDE's prediction largely differs from ground truth.

Figure 15–18 shows more qualitative continuum field reconstruction results with different number of sensing locations ($N \in \{256, 128, 64\}$) on $NS_{\nu 1e-3}$, $NS_{\nu 1e-5}$, *Shallow-Water* and *Sea Surface Temperature* benchmark.

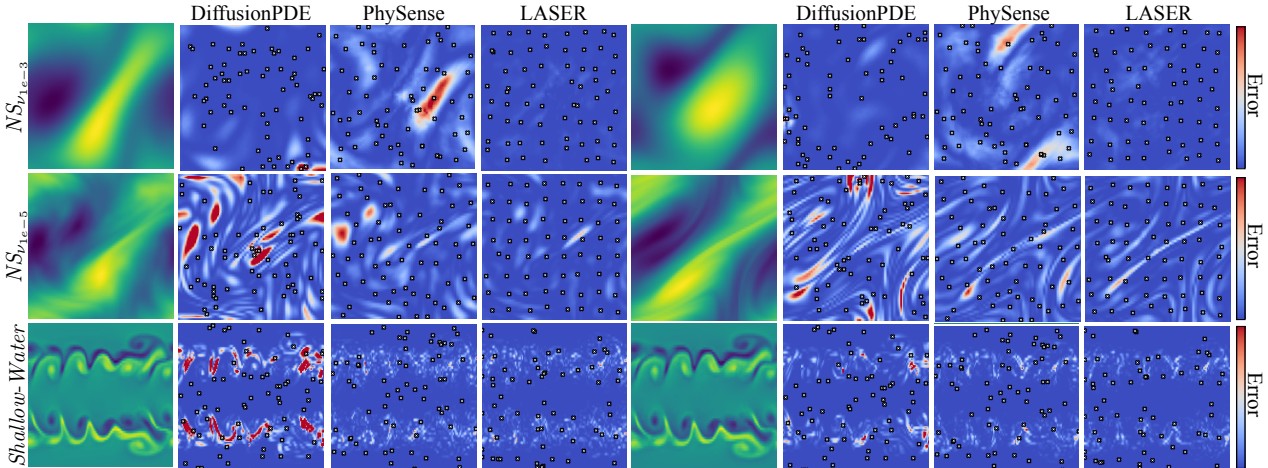

*Figure 13.* **Qualitative showcases of active sensing methods on $NS_{\nu 1e-3}$, $NS_{\nu 1e-5}$ and *Shallow-Water*.** We evaluate different placement strategies under extreme sparsity ($N = 64$).

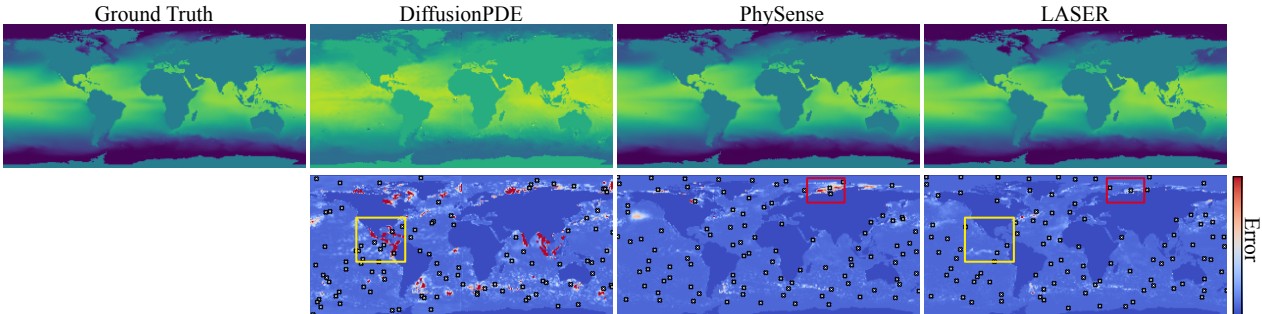

*Figure 14.* **Qualitative showcases of active sensing methods on *Sea Surface Temperature*.** We evaluate different placement strategies under extreme sparsity ($N = 100$).

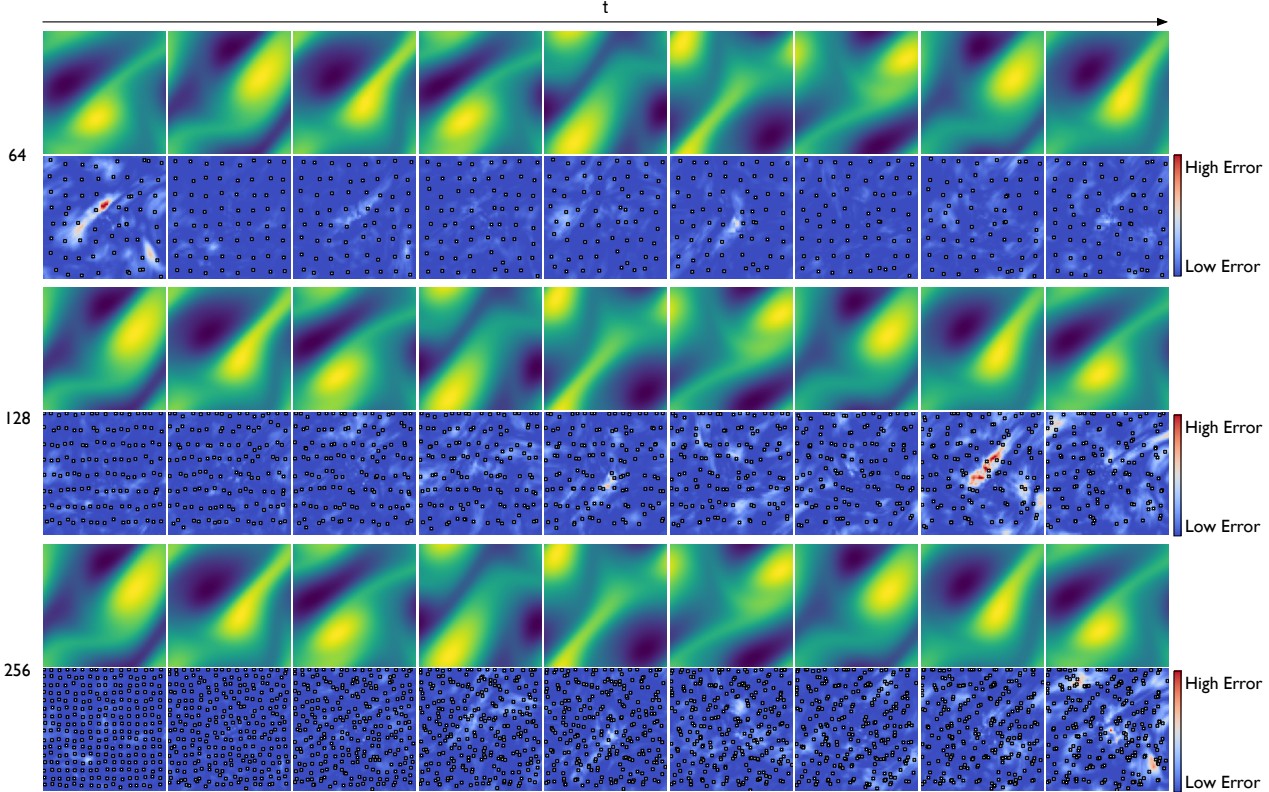

*Figure 15.* Showcasses of continuum field reconstruction on $NS_{\nu 1e-3}$ across different sparse observations by LASER.

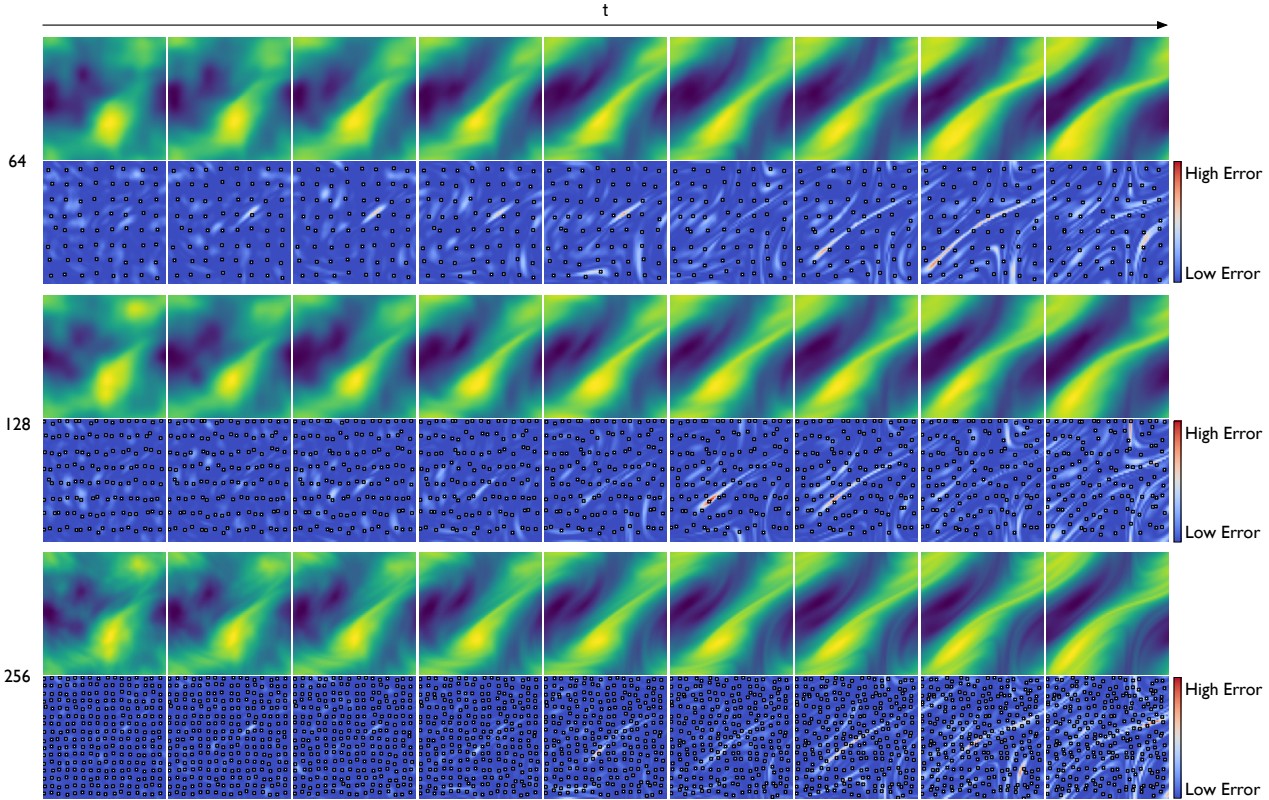

*Figure 16.* Showcasses of continuum field reconstruction on $NS_{\nu 1e-5}$ across different sparse observations by LASER.

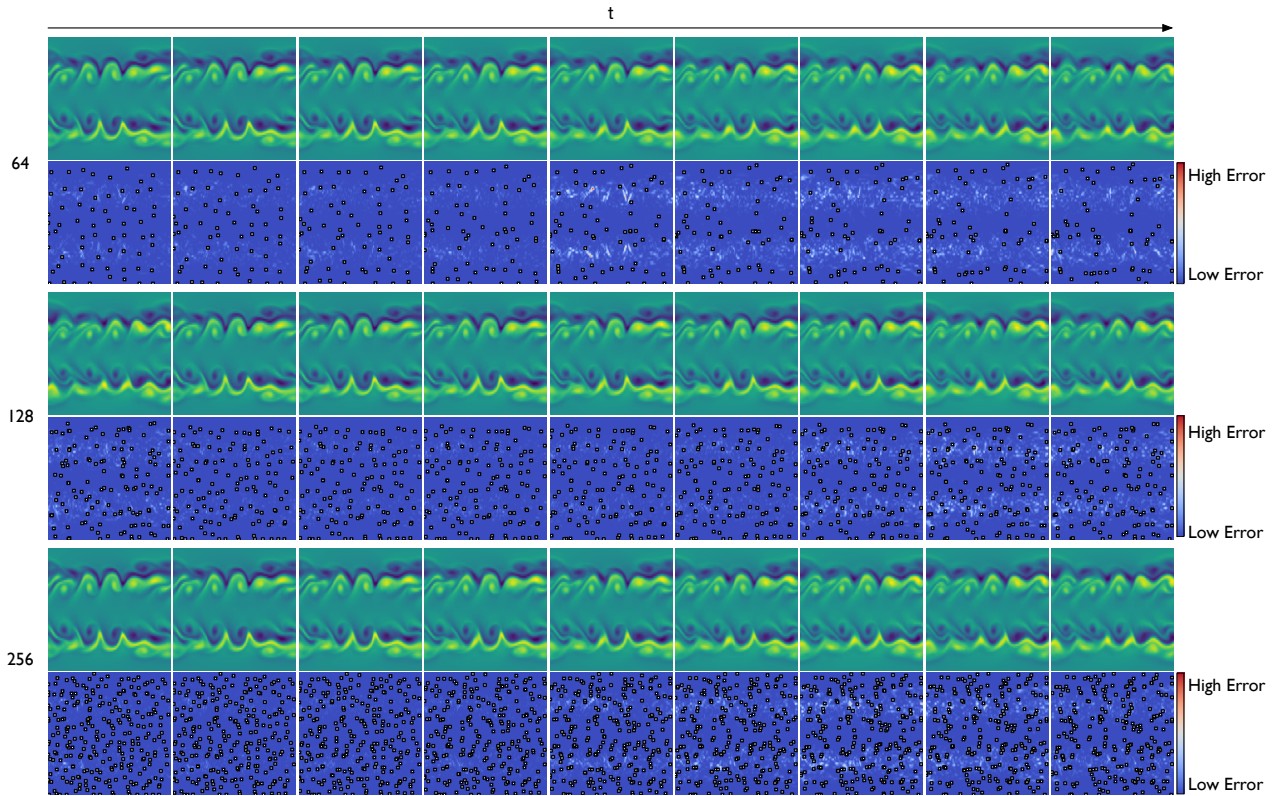

*Figure 17.* Showcasses of continuum field reconstruction on *Shallow-Water* across different sparse observations by LASER.

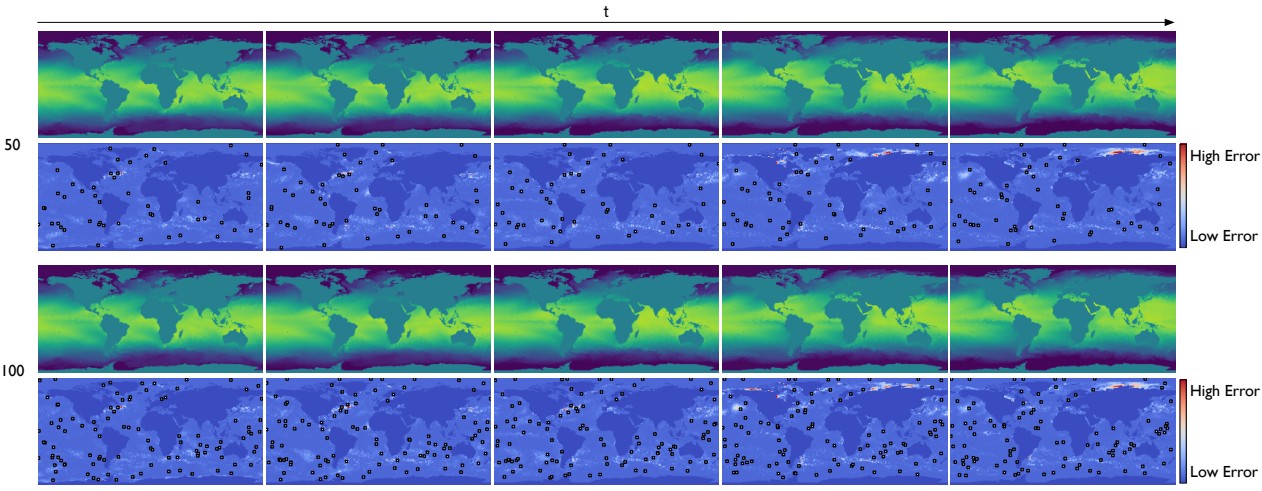

*Figure 18.* Showcasses of continuum field reconstruction on *Sea Surface Temperature* across different sparse observations by LASER.

