# OpenReview forum: "LASER: Learning Active Sensing for Continuum Field Reconstruction"
_ICML.cc/2026/Conference — ICML 2026 spotlight_

### Official Review · Reviewer_o5Er · 2026-03-11

**Soundness:** 3
**Presentation:** 3
**Significance:** 3
**Originality:** 2
**Overall Recommendation:** 3
**Confidence:** 4

**Summary:**

The proposed method, LASER, treats active sensing as a Partially Observable Markov Decision Process (POMDP). By utilizing a latent world model of continuum fields (i.e., a transformer-based conditional diffusion model), LASER replicates physical dynamics and generates intrinsic rewards, allowing a reinforcement learning agent (GRPO) to explore what-if scenarios in a latent space. This approach enables the system to move sensors toward high-information areas by anticipating future states beyond the immediate data. Extensive experiments demonstrate that LASER achieves superior fidelity across diverse physical domains.

**Compliance With Llm Reviewing Policy:**

Affirmed.

**Key Questions For Authors:**

The justification for formulating active sensing as a Partially Observable Markov Decision Process (POMDP) requires further elaboration. Specifically, the mechanism by which the Group Relative Policy Optimization (GRPO) agent navigates the exploration-exploitation trade-off to minimize reconstruction error remains opaque. To strengthen this claim, the authors should provide qualitative visualizations and quantitative trajectory-level analyses that illustrate the agent's decision-making logic over time.

Furthermore, the necessity of a separate reinforcement learning module like GRPO, which often presents a significant training bottleneck, needs rigorous validation. Given that the world model utilizes a conditional diffusion model, it is worth investigating whether the intrinsic uncertainty or entropy derived from the model's posterior reconstructions guided by observation history would suffice for decision-making. Could a simpler, uncertainty-guided heuristic eliminate the need for an independent RL training phase entirely?

**Limitations:**

While the framework is technically sound, its contributions appear somewhat incremental, as the integration of world models with reinforcement learning for active sensing has been previously explored. The decision to model the environment as a POMDP warrants a more robust justification; specifically, the authors should clarify the tangible benefits this complexity provides over simpler formulations.

Furthermore, the role of GRPO requires more rigorous validation. A primary concern is that the training of the GRPO module presents a significant bottleneck, particularly in scientific discovery contexts where high-fidelity annotated data is scarce. Relying on such data-intensive supervision may limit the framework's practical applicability in real-world scenarios where labels are difficult to obtain.

**Strengths And Weaknesses:**

Strengths:

The submission is technically sound and addresses a significant challenge in scientific discovery: high-fidelity reconstruction of continuum physical fields under sparse sensing. The proposed architecture, integrating a conditional diffusion model for state dynamics with a Group Relative Policy Optimization (GRPO) agent, is a logical and well-reasoned approach to active sensing. Furthermore, the paper is well-structured, making the complex interplay between the world model and the RL policy easy to follow. Experimental results further supports the validity of the proposed framework.

Weaknesses:

Despite its technical execution, the framework is somewhat incremental. The integration of world models with RL agents for sensor placement has been explored in prior literature, and this work follows a similar trajectory without a distinct departure from existing paradigms.

A more critical concern lies in the practical applicability of the LASER framework. The GRPO module requires a substantial amount of supervised or annotated data for every evolving state and potential sensor location. In many real-world scientific discovery settings, such high-resolution ground truth data is prohibitively difficult or expensive to obtain. This creates a significant training bottleneck and limits the model's scalability across different domains. In these environments, unsupervised or self-supervised approaches are generally preferred to mitigate data scarcity. The authors should clarify how the framework might adapt to scenarios where domain-specific supervised data is limited, as the current requirement for dense labels may hinder its adoption in the very scientific fields it aims to serve.

---

> ### Author Rebuttal · Authors · 2026-03-31
>
> We appreciate the reviewer's valuable comments.
> ### 1. Why POMDP + RL?
> >Limitation 1. Justification of POMDP + RL
>
> **Sensor placement is a sequential decision-making problem under partial observations.** Reconstruction is highly sensitive to observation locations (Ma et al., 2025), motivating *optimization* over fixed/random layouts. Sensor choices determines acquired information, inducing long-horizon dependencies. In our POMDP, the policy selects observation locations, and latent dynamics evolve via the learned world model.
>
> **Simple formulations are insufficient for non-stationary physical systems.** Static methods (e.g., PhySense) learn a single "optimal" layout that cannot adapt to evolving dynamics (Tables 4 & 5). Heuristics(e.g entropy-based) prioritize local high-variance regions and ignore transition costs, leading to suboptimal decisions (see response to Q2).
> >Q1. Trajectory-level analyses.
>
> We provide visualizations in Fig. 13–16 in Appendix and videos (Videos 4-5) at https://anonymous82711.github.io/laser, including sensor placements, reconstructions, and error maps.
>
> To quantify policy behavior, we analyze trajectories on $\text{NS}_{\nu 1e-3}$ ($N=256$) using two metrics:
> - $MSE_t$: per-timestep reconstruction error.
> - Mean error-gradient at sensor locations: average magnitude of the spatial gradient of the error field at the sensor coordinates, measuring local irregularity.
>
> As seen in Fig. 3-4 in the link and the following table, LASER's improvement is persistent on $MSE_t$. The learned policy also yields lower error-gradients than random layouts, indicating reduced spatial inconsistency in the error field rather than targeting high-gradient regions.
>
> Metric(x1e-4)|Random Sampling|LASER|Reduction
> -|-|-|-
> In-t MSE|1.67|0.781|53.2%
> Out-t MSE|4.99|4.22|15.4%
> Average MSE|4.16|3.02|27.4%
> >W1. Prior works on world models with RL for sensor placement.
>
> We are not aware of prior work *"world models + RL for sensor placement"*. If such work exists, we would appreciate references and will include careful discussion.
>
> To our knowledge, this is the first work on **closed-loop, online sensor placement via a latent world model with RL**, formulated as a POMDP over latent physical states for sequential and instance-specific adaptation from partial observations.
>
> Prior works fall into distinct categories:
> - Sensor placement optimized during training but fixed at inference (e.g., [1]), lacking adaptability to instance-specific dynamics;
> - RL control or state estimation with physical models (e.g., [2][3]), where sensing is not part of the decision process.
>
> None of these methods perform closed-loop decision-making over sensor locations. The distinction is critical: Our method adaptively select sensing actions based on evolving field dynamics and partial observations, unlike static or precomputed layouts. Ablations (see response to Q2) show heuristics underperform, highlighting the benefit of online, adaptive sensing.
>
> [1] MCAS. Zhou et al. [2] PIR. Fent et al. [3] RL-ROE. Mowlavi et al.
> ### 2. Data cost
> >W2 & Limitation 2. Data need.
>
> While RL training is often data-intensive, **this is not the case in our setting**: LASER introduces no additional data requirements beyond standard supervised baselines.
> - After training the world model $\phi$ on offline data, it is frozen. GRPO optimizes $\pi_\theta$ by varying spatial sampling coordinates and querying the latent representations of the world model, without new data collection.
> - Rewards are computed from the reconstruction error. Evaluating different sensing strategies is therefore reduced to querying different points on a pre-computed latent manifold, rather than executing $G$ separate simulations.
>
> Notably, all continuum field reconstruction methods require high-resolution data of the physical field. Our world model is trained on the same dataset as the baselines.
> ### 3. Heuristic baselines
> >W3 & Q2. Uncertainty-based heuristics
>
> We thank the reviewer for this suggestion. We implement three sensing heuristics:
> - Entropy-based: Selects locations with the highest predictive uncertainty (posterior entropy) derived from the world model's latent reconstruction.
> - Error-based: Selects locations with the high reconstruction error from the previous time step.
> - Dynamics-based: Selects locations with the largest temporal changes.
>
> In all cases, sampling probabilities are proportional to heuristic scores. We evaluate these baselines on NS$_{1e-3}$.
> ||256(x1e-3)|128(x1e-3)|64(x1e-3)
> -|-|-|-
> Entropy-based|1.387|7.467|52.50
> Error-based|0.869|4.098|26.61
> Dynamics-based|0.898|5.251|28.58
> LASER|0.096|0.149|0.169
>
> **Key findings:** As shown in the table, all three baselines significantly underperform, with errors often 10$\times$ to 100$\times$ higher than LASER, as they optimize instantaneous signals and ignore temporal coupling and lack "proactive" logic to position sensors.
>
> **Visual evidence:** Qualitative comparisons are in Videos 1-4 in the link.

---

> > ### Author Rebuttal · Reviewer_o5Er · 2026-04-04
> >
> > While I appreciate the authors' detailed rebuttal, my concerns regarding the technical novelty remain. The proposed framework appears to be an integration of established modules tailored to a specific application, rather than a fundamental algorithmic contribution. Furthermore, for a venue such as ICML, a more rigorous theoretical grounding is necessary to support the empirical findings. Consequently, I am maintaining my original score.

---

> > > ### Author Response · Authors · 2026-04-04
> > >
> > > We thank the reviewer for the continued engagement and address the remaining concerns.
> > >
> > > ### 1. Information-theoretic justification: adaptive sensing strictly outperforms static sensing.
> > >
> > > We appreciate that theoretical analysis can strengthen empirical work. We provide the following theoretical justification.
> > >
> > > **Setup.** Consider a spatiotemporal field $u_t: \Omega \to \mathbb{R}^C$ over $t=1,\ldots,T$. At each step, $N$ sensors at locations $X_t \subset \Omega$ yield observations $o_t$. A *static* strategy fixes $X_t = X^{s}$ for all $t$; an *adaptive* strategy selects $X_t = \pi(o_{<t})$ based on observation history. The per-step reconstruction error is $MSE_t(X_t) = \mathbb{E} [\frac{1}{|\Omega|}\int_\Omega \|u_t - \hat{u}_t\|^2 dx]$.
> > >
> > > **Proposition.** Let $\mathcal{E}^{s\*}$ and $\mathcal{E}^{a\*}$ denote the minimum cumulative error $\sum_t MSE_t$ under the optimal static and adaptive strategies. Then $\mathcal{E}^{a\*} \leq \mathcal{E}^{s\*}$, with **strict inequality** whenever the per-step optimal placement $X_t^\* = \arg\min_X MSE_t(X)$ varies across timesteps.
> > >
> > > **Proof.**
> > >
> > > *(i) Weak inequality.* The adaptive policies $\Pi^{a}$ contains all static policies: any $X^{s}$ can be implemented by $\pi$ that ignores inputs and outputs $X^{s}$. Thus $\mathcal{E}^{a\*} = \min_{\pi \in \Pi^{a}} \sum_t MSE_t(\pi) \leq \min_{X^{s} \in \Pi^{s}} \sum_t MSE_t(X^{s}) = \mathcal{E}^{s\*}$.
> > >
> > > *(ii) Strict inequality for non-stationary fields.* An oracle adaptive policy selecting $X_t^\*$ at each step achieves $\sum_t MSE_t(X_t^\*)$. The optimal static placement $X^{s\*} = \arg\min_{X} \sum_t MSE_t(X)$ is a single compromise layout, so $MSE_t(X_t^\*) \leq MSE_t(X^{s\*})$ for all $t$, with equality only if $X^{s\*}$ is also optimal at step $t$. When $\exists\, t_1, t_2$ with $X_{t_1}^\* \neq X_{t_2}^\*$, no single $X^{s\*}$ can be optimal at every $t$, so the inequality is strict for at least one $t$, yielding $\mathcal{E}^{a\*} \leq \sum_t MSE_t(X_t^\*) < \sum_t MSE_t(X^{s\*}) = \mathcal{E}^{s\*}$. $\square$
> > >
> > > **Corollary.** The gap $\mathcal{E}^{s\*} - \mathcal{E}^{a\*}$ grows with the degree of non-stationarity (i.e., how much $X_t^\*$ varies across $t$).
> > >
> > > LASER approximates $\pi^\*$ through learned components: the world model evaluates reconstruction quality and forecasts future states ($\hat z_{t+1}$), and the RL policy is trained to minimize cumulative MSE and enabling anticipation of $X_{t+1}^\*$. The predictive reward horizon $H$ (Eq. 12, Table 8) controls how closely the policy tracks the time-varying optimal placement. We will include this theoretical analysis in the revision.
> > >
> > > ### 2. On novelty: "integration of established modules".
> > >
> > > We would like to repeat that we are not aware of any prior work that addresses the **same** problem — adaptive sensor placement for continuum physical field reconstruction, **which is a critical problem that exists in physics application**. We would welcome the reviewer pointing to a specific prior work that achieves what LASER does. Absent such a reference, we maintain that the contribution is novel and significant.
> > >
> > > LASER's contribution is not any individual module, but:
> > >
> > > 1. **The problem formulation** — formalizing continuum field active sensing as a POMDP, which has not been done before.
> > > 2. **The proactive sensing mechanism** — conditioning the policy on predicted future latent states with a newly proposed latent world model structure, enabling proactive placement.
> > > 3. **Empirical validation that the full pipeline is necessary** — the heuristic baselines confirm that simpler alternatives substantially underperform, demonstrating that the "integration" produces emergent capabilities that individual components cannot achieve.
> > >
> > > We note that ICML has published works[1-5] whose primary contributions are methodological frameworks through thorough experiments — not all accepted papers introduce novel neural network modules(e.g., new attention mechanisms or RNN architectures), and **contributions to specific scientific domains are equally central to the evaluation criteria**. Our contributions include: (i) active sensing formulation with sparse observations in scientific fields, (ii) a novel latent world model that explicitly connects to the POMDP formulation via graphical modeling, and (iii) comprehensive empirical validation, including ablations, strong baselines, and additional heuristic alternatives evaluated in the rebuttal.
> > >
> > > [1] Unisolver: PDE-Conditional Transformers Towards Universal Neural PDE Solvers. ICML 2025
> > >
> > > [2] A Variational Framework for Improving Naturalness in Generative Spoken Language Models. ICML 2025
> > >
> > > [3] OneForecast: A Universal Framework for Global and Regional Weather Forecasting. ICML 2025
> > >
> > > [4] AutoML-Agent: A Multi-Agent LLM Framework for Full-Pipeline AutoML. ICML 2025
> > >
> > > [5] Towards a Self-contained Data-driven Global Weather Forecasting Framework. ICML 2024
> > >
> > > [6] TimeSiam: A Pre-Training Framework for Siamese Time-Series Modeling. ICML 2024

---

### Official Review · Reviewer_SUts · 2026-03-12

**Soundness:** 3
**Presentation:** 4
**Significance:** 3
**Originality:** 3
**Overall Recommendation:** 5
**Confidence:** 3

**Summary:**

The authors proposed to address a key issue in sparse physical sensing: fixed or offline-optimized sensor layouts are often mismatched to non-stationary field dynamics. The article analyses the aspect of closed-loop coupling between latent prediction and sensing control by formulating active sensing as a POMDP and using a latent world model for forward prediction and reward evaluation. A policy is trained to move sensors toward informative regions. LASER usually improves both rollout quality and online reconstruction over static baselines, and often over PhySense and a PPO variant, especially under higher sparsity and on the SST benchmark. The ablations confirm the two main design choices including dynamic quality filtering and longer-horizon predictive rewards improve performance, while history in the latent dynamics model helps out-of-distribution rollout.

**Compliance With Llm Reviewing Policy:**

Affirmed.

**Key Questions For Authors:**

N/A

**Limitations:**

See the weaknesses listed above.

**Strengths And Weaknesses:**

Strengthes
•	Formulation of continuum-field sensing as a latent POMDP, rather than treating reconstruction and placement as decoupled stages. This is a meaningful step beyond fixed-layout reconstruction and offline sensor optimization.
•	The world-model design is meaningful, by integrating sparse observation encoder, latent dynamics predictor, and decoder-based reward evaluation into one predictive surrogate for planning.
•	A proactive policy architecture that conditions on predicted future latent states instead of only current observations, which is the paper’s most interesting technical idea.
Major weaknesses
•	The method combines largely familiar ingredients latent encoder/decoder reconstruction, GRU memory, diffusion-based latent prediction, transformer policy, and GRPO-style RL into a new pipeline.
•	The evaluation could be enhanced. The introduction motivates sparse, irregular, noisy, and partial measurements, but the experiments mainly study sparsity and temporal extrapolation. There is little evidence on noise, sensor dropout, or observational datasets. Glorys also has higher resolution data that could be tested.
•	The method is heavily dependent on the learned world model. How policy quality degrades when dynamics prediction is imperfect?
Minor weaknesses
•	Pseudocode/reward inconsistency. Algorithm 1 says it computes lookahead reward (r^{lookahead}), but the filtering condition and stored rollout tuple use (r^g_t).
•	The mapping from group action to per-sensor action should be defined more clearly.
•	A discrete candidate set (D) for possible sensor positions is defined in section 3, while the implementation later uses continuous displacements.  Further details should be provided early in the problem definition.
•	Several presentation errors remain. For example, “Perfrom” in Algorithm 1, “Figire 4” in the ablation section, “paramerters” in Table 10, and “borrrow” in the DiffusionPDE appendix description.
•	The appendix states that DiffusionPDE takes tens to hundreds of seconds per frame while LASER takes about 0.5 seconds, but this important efficiency claim is not integrated into the main experimental section with sufficient evidence.

---

> ### Author Rebuttal · Authors · 2026-03-31
>
> We thank the reviewer for the valuable comments.
> >W1. The method combines largely familiar ingredients into a new pipeline.
>
> We respectfully argue that the contribution goes beyond combining standard components:
>
> **Formulation novelty.** LASER is the first to cast active sensing for continuum fields as a POMDP with a latent world model. Prior works assume fixed or offline-optimized layouts. Our key contribution is enabling **proactive sensing** by conditioning decisions on predicted latent states (Eq. 3), shifting from reactive to proactive strategies.
>
> **Necessity of active sensing framework.** Our heuristic baselines (see Reviewer o5Er, Q2) show that simpler sensing heuristics (e.g., entropy/error-based) underperform significantly, indicating that gains arise from the **integration** of world modeling and RL, not individual components.
>
> As summarized in Table 1, no prior method jointly supports sparse observations, forward prediction, placement optimization, and **closed-loop active sensing**; LASER provides this unified framework.
> >W2. Evaluation lacks noise, sensor dropout.
>
> We agree this is an important evaluation dimension. We conduct additional experiments below:
>
> - **Noisy observations.** We added Gaussian noise to sensor measurements during inference on NS$_{1e-3}$ (N=256): LASER maintains consistent performance under moderate noise, with acceptable degradation.
>
> ||AROMA($\times 1e-3$)|||LASER($\times 1e-3$)|||
> |-|-|-|-|-|-|-|
> |Noise Scale|In-t|Out-t|Avg MSE|In-t|Out-t|Avg MSE|
> |0|2.19|3.25|2.72|0.099|0.482|0.304|
> |0.01|2.20|3.24|2.72|0.102|0.651|0.377|
> |0.05|2.33|3.30|2.81|0.187|0.735|0.461|
>
> - **Sensor dropout.** We randomly drop $p\%$ of sensors at each timestep during inference to simulate sensor failure: despite missing observations, LASER maintains stable performance, showing robustness to partial and irregular sensing:
>
> ||AROMA($\times 1e-3$)|||LASER($\times 1e-3$)|||
> |-|-|-|-|-|-|-|
> |Drop Prob.|In-t|Out-t|Avg|In-t|Out-t|Avg|
> |0%|2.19|3.25|2.72|0.099|0.483|0.304|
> |3%|2.25|3.32|2.78|0.102|0.430|0.266|
> |6%|2.41|3.39|2.90|0.119|0.667|0.393|
> |12%|2.61|3.60|3.11|0.129|0.769|0.449|
> |25%|3.17|4.04|3.61|0.139|0.653|0.396|
>
> These results demonstrate that the learned policy is not only effective under sparsity, but also robust to noise and dynamic sensor availability.
> >W3. How does policy quality degrade when dynamics prediction is imperfect?
>
> We thank the reviewer for this important question. In our framework, the dynamics prediction affects policy only through the **predicted latent state** $\hat{z}_{t+1}$ used for action selection. Since LASER makes **per-step decisions**, prediction error only influence the next sensing action, rather than propagating through multi-step rollouts. At the next time step, the latent state is **re-inferred** from new observations, so errors do not accumulate over time.
>
> Furthermore, iterative co-training of the world model and policy further improves performance (see response to Reviewer 1x42, W1 & W2). This suggests that improving the latent world model directly strengthens the policy, while the decoupled formulation remains robust even when the dynamics model is imperfect.
> >W4 & W7. Pseudocode inconsistency — $r^{\text{lookahead}}$ vs $r^g_t$. Typos.
>
> Thank you for catching these typos. All typos will be corrected in the revision.
> >W5. The mapping from group action to per-sensor action should be defined more clearly.
>
> Please see response to Reviewer xwSg Q1.
> >W6. $\mathcal{D}$ vs continuous displacements
>
> When optimizing sensing policy, sparse observations at continuous sensor coordinates are obtained via bilinear interpolation (input side), while reconstruction loss is evaluated at grid points (output side). We will clarify this notation in the revision.
> >W8. Efficiency analysis. DiffusionPDE vs. LASER.
>
> We agree that efficiency is a critical aspect and will add explicit measurements to the main experimental section. Specifically, we benchmark inference time(including sensor placement and reconstruction) on NS$_{1e-3}$ under varying sensing budgets (256/128/64 sensors). The results show a consistent and substantial gap:
>
> * DiffusionPDE: 445.4s / 442.9s / 448.7s per frame
> * PhySense: offline sensor placement optimization on training set, 0.3 hours.
> * LASER ($\phi$, $\theta$): 16.8ms / 16.9ms / 16.8ms per frame
>
> DiffusionPDE performs iterative denoising over the full physical field, requiring many diffusion steps to ensure reconstruction quality, even when sensor locations are fixed. Instead, our method directly reconstructs the field in a single forward pass conditioned on observations, leading to lower latency. We will move this comparison from the appendix to the main paper and include the above quantitative results to better support the efficiency claim.

---

> > ### Author Rebuttal · Reviewer_SUts · 2026-04-06
> >
> > We thank the authors for their good efforts. The evaluation under noise and sensor dropout has been partially addressed, but these appear to be inference-time perturbations only. Another remaining concern is that inaccurate dynamics may degrade performance, since the policy relies on the latent model. Additional tests comparing policies trained with stronger versus weaker world models would be useful.

---

> > > ### Author Response · Authors · 2026-04-07
> > >
> > > We thank the reviewer for the helpful suggestions and have conducted additional experiments to address them more directly.
> > >
> > > **(1) Training-time noise and sensor dropout**
> > > We have extended experiments on NS$_{1e-3}$ by incorporating training-time perturbations: (i) additive Gaussian noise (scale: 1e-2) on sparse observations, and (ii) random sensor dropout with rates $p$\% $\in$ {3\%, 6\%, 12\%}. The policy is trained under these stochastic corruptions and **evaluated under clean (no perturbation) conditions**. We report the average reconstruction error over the full sequences.
> > >
> > > As shown below, training with perturbations degrades performance, as expected, due to the distribution shift between training and test conditions. Noise Injection (0.358) leads to larger errors than Sensor Dropout (0.315), compared to the Clean setting (0.304). This suggests the policy is more robust to partial observability but more sensitive to additive noise. Overall, the policy does not rely on idealized observations and remains stable under realistic corruptions.
> > >
> > > | Train Corruption | Avg($\times1e-3$)|
> > > | ---------------- | ----- |
> > > | Noise Injection  | 0.358 |
> > > | Sensor Dropout   | 0.315 |
> > > | Clean            | 0.304 |
> > >
> > > **(2) Sensitivity to latent world model performance**
> > > We agree that the quality of the latent world model is an important factor. To investigate this, we train policies on top of world models with varying fidelity, including weaker models trained with reduced optimization (e.g., 80\% training steps) and stronger world models (100\% training steps) on the NS$_{1e-3}$ dataset and report average reconstruction error.
> > >
> > > As expected, better world models lead to improved downstream performance. However, we find that even with weaker models, the learned policy still outperforms static sensing baselines, indicating robustness to moderate model inaccuracies. This suggests that active sensing remains beneficial even under some degree of model degradation.
> > >
> > > | Capability   | PhySense($\times1e-3$) | LASER($\times1e-3$) |
> > > | ------------ | -------- | ----- |
> > > | Weak 80\%-steps    | 0.452    | 0.387 |
> > > | Strong 100\%-steps | 0.376    | 0.304 |
> > >
> > > In addition, we include results with **iterative training of the world model and policy** (see responses to **Reviewer 1x42's W1&W2**). As a result, it effectively yields a stronger world model (with improved predictive accuracy), which in turn leads to better policy performance. This provides additional evidence that improved model fidelity consistently brings in improved downstream performance.
> > >
> > > These new experiments demonstrate that (i) training-time perturbations cause only moderate performance degradation, and (ii) the proposed active sensing strategy remains effective under imperfect dynamics,  and achieves further gains from stronger world models.
> > >
> > > We will include the above results in the revised paper.

---

### Official Review · Reviewer_xwSg · 2026-03-12

**Soundness:** 3
**Presentation:** 3
**Significance:** 2
**Originality:** 3
**Overall Recommendation:** 4
**Confidence:** 4

**Summary:**

The paper presents a model-based reinforcement learning algorithm for adaptive sensor placement in the monitoring of continuum dynamical processes, such as those governed by the Navier-Stokes equations. The proposed approach decouples model learning from control: a world model is trained offline, while sensor placement is formulated as a Partially Observable Markov Decision Process (POMDP). Given observations from N sensors at fixed locations, a transformer-based encoder maps the sensor readings into a latent representation, and a learned forward model predicts the latent embedding of the subsequent time step. This predicted embedding is then provided as input to a policy network, which outputs the next sensor positions. The policy is trained using Group Relative Policy Optimization (GRPO). The method is evaluated on three benchmark continuum processes and is reported to outperform several state-of-the-art baselines.

**Compliance With Llm Reviewing Policy:**

Affirmed.

**Final Justification:**

The paper introduces sensor placement as a PoMDP problem and also reformulates GRPO to be applicable in this continuous control setting. The rebuttal could address all my concerns regarding the architecture and the ablation. In addition, the authors provide new results that show a clearer advantage of their method using iterative training of the world model and the policy. Due to the interesting problem statement, I think the paper should be published.

**Key Questions For Authors:**

- Permutation invariance: It is unclear whether the proposed architecture is truly permutation-invariant with respect to sensor ordering. Specifically, is the MLP at the end of the encoder applied independently per sensor token, or does it operate on a joint embedding of all sensor tokens? If the latter, the model may not be invariant to permutations in sensor order, which would be an important limitation to clarify or address.
- Computational cost: Could the authors provide a more detailed account of the computational cost of their experiments? This should include wall-clock training time, hardware used, and any relevant efficiency comparisons to baselines.
Learning curves: Please include learning curves for the main experiments. These would help assess the convergence behavior of the GRPO-based training procedure and give readers a sense of the variance across random seeds, both of which are important for evaluating the reliability of the reported results.
- Notation in Eq. 4: It appears that p(u|z) in Equation 4 should read p(u|z, x), as the policy would naturally condition on both the latent state z and the current observation x. Could the authors clarify whether this is a typo or an intentional modeling choice?
Relation to Dreamer: The mathematical formulation of the world model learning framework bears a strong resemblance to the Dreamer architecture (Hafner et al.). Could the authors explicitly discuss the similarities and differences? In particular, it would be valuable to understand what novelty is claimed relative to Dreamer, and whether any components are directly borrowed or adapted from it.
- Short-horizon optimization and sensor count: The paper appears to optimize only over very short horizons, yet little justification is provided for this design choice. Could the authors clarify how many time steps constitute a single episode? Furthermore, an ablation comparing short-horizon optimization against a standard long-horizon formulation with discounting would greatly strengthen the empirical analysis. More broadly, the short-horizon setting may be particularly limiting when the number of available sensors is small. With fewer sensors (e.g., fewer than 10), the agent must make more consequential placement decisions, which intuitively requires longer-horizon planning to be effective, making the underlying MDP substantially harder to solve. It would therefore be valuable to see a more detailed ablation of the number of sensors, and in particular an analysis of how the choice of planning horizon interacts with sensor count. Does performance degrade more sharply with fewer sensors, and does a longer horizon help in that regime?

**Limitations:**

Limitations in terms of the limited horizon length and the number of sensors (does it work for a small amount of sensors?) should be discussed in more detail.

**Strengths And Weaknesses:**

Strengths:

- The formulation of adaptive sensor placement as a POMDP is well-motivated and provides a principled framework for handling partial observability in continuum process monitoring.
- The proposed architecture is flexible and practically appealing: by using an encoder-decoder design, the model can naturally accommodate a variable number of sensors without architectural modifications.
- The empirical results are convincing, with the proposed method consistently outperforming state-of-the-art baselines across all three evaluated benchmarks.


Weaknesses:

- The algorithm is evaluated only in short-horizon settings. It remains unclear how the method scales to longer planning horizons, where compounding prediction errors in the learned forward model may significantly degrade performance.
- Several algorithmic design choices lack sufficient ablation. For example, [insert specific choices here, e.g., the choice of GRPO over other policy optimization methods, the transformer architecture, etc.] — it is difficult to assess the individual contribution of each component to the overall performance.
- The paper lacks important implementation details necessary for reproducibility and computational assessment. Specifically, the authors should report: training computation time, the number of rollouts per group and groups per batch in GRPO, and learning curves to assess convergence behavior.

---

> ### Author Rebuttal · Authors · 2026-03-31
>
> We thank the reviewer for the valuable comments.
> > W1. Evaluated in short-horizon settings.
>
> **Multi-step planning**. We believe this concern stems from a formulation mismatch. LASER is not a planning-based methods (unlike Dreamer/MPC mehtods). It learns a **one-step** policy conditioned on the predicted latent $\hat{z}_{t+1}$ and does not perform multi-step planning over *action* sequences in training or testing(see Q3.2, Q4.1).
>
> **Multi-step reward.** The world model is only used to compute a predictive reward over a window $H$ to evaluate the current action. Table 8 shows an ablation of $H$.
>
> >W2. Ablation. GRPO vs. other policy optimization methods.
>
> We have included GRPO vs. PPO in Table 4, where GRPO shows modest but consistent gains across datasets and sparsity levels. For architecture design, the latent world model is **backbone-agnostic** and compatible with other optimization methods (see responses to Reviewer 1x42, W2). We will further include scaling ablations (model size / capacity) in the revision.
> >W3. Missing details.
>
> Implementation details are in Appendix B and E, with hyperparameters in Tables 9–10. We use GRPO group size $G=8$ and batch size 512 (64 groups). Training time is discussed in Q2, and training curves (1M steps) are in Fig.1–2 at https://anonymous82711.github.io/laser/.
> >**Q1.** Permutation invariance.
>
> The policy $\pi_\theta$ is **permutation-equivariant** w.r.t the sensing ordering and supports variable sensor sets. The policy consists of: queries from observations, cross-attention to $\hat z_{t+1}$ (invariant), self-attention across sensors (equivariant), and a MLP that outputs $a_t^i$ independently. Each component is permutation-equivariant (or invariant), so permuting inputs induces the same permutation on outputs. *Note, the MLP mentioned by the reviewer belongs to $\pi_\theta$, not the world model $\phi$.*
> >Q2. Training cost.
>
> We report training hours and per-optimization runtime on NS$_{1e-3}$ with 256 sensors, with 4 historical steps for $p^{dyn}$. PhySense optimizes a globally fixed sensor layout on top of pretrained LASER($\phi$). All experiments use a single RTX 3090 (Appendix E). Inference cost is discussed in responses to SUts, W8.
> ||Method|Hours/h|per-optimization/ms
> |-|-|-|-|
> |Pretraining|AROMA (Enc-Dec)|15.1|44.8|
> ||AROMA (Dyn)|1.31|7.1|
> ||DiffusionPDE|8.98|44.5|
> ||LASER($\phi$) (Enc-Dec)|15.1|44.8|
> ||LASER($\phi$) (Dyn)|1.98|35.2|
> |Sensor Opt.|PhySense|0.31|27.5|
> ||LASER($\theta$)|5.57|55.2|
>
> >Q3.1. Eq.4: $p(u|z)$ should be $p(u|z, x)$, as the *policy* ...
>
> We clarify that Eq. 4 defines the **latent world model** objective, rather than the policy objective. The continuum field decoder reconstruct $\hat u_t(x) = p_\phi^{dec}(\cdot | z_t, x), \forall x \in \Omega$. In contrast, Eq. 3 defines the policy $\pi_\theta(\cdot | \hat z_{t+1}, o_t)$, which conditions on the predicted latent and current observations.
> > Q3.2. Relation to Dreamer.
>
> While both LASER and Dreamer adopt latent world models to capture temporal dependencies, their roles are fundamentally different:
> - **Action-conditioned transitions vs. Passive physics**: Dreamer's world model learns $p(s_{t+1}|s_t,a_t)$, where the actions affect state. LASER($\phi$) models passive dynamics and $\pi_\theta$ does not affect the physics transition $u_t \to u_{t+1}$, but only controls sensor placement.
> - **Action planning vs. Proactive action context**: Dreamer performs rollouts in the latent space for long-horizon planning over $a_{\tau:\tau+H}\sim \pi(s_t)$. LASER uses predicted latent ($\hat z_{t+1}$) as a "lookahead" physical context of the field's future manifold, which informs the sensing policy $a_t \sim \pi(\hat z_{t+1},o_t)$ of where information is likely to emerge.
> - **Value learning vs. Predictive reward**: Dreamer predicts the rewards and uses it to learn the value function. LASER uses predictive reward (Eq. 12) to evaluate the one-step action and optimizes with GRPO.
>
> >Q4.1. Episode length.
>
> We follow **standard** data lengths without artificial truncation (20–49 steps, Appendix C). For simulated PDE datasets, $\phi$ and $\pi_\theta$ are trained on the first half of each trajectory (In-t) and evaluated on the full sequences, to test generalization. LASER does not perform long-horizon planning (see W1, Q3.2); its “horizon” refers to the reward estimation window, not an action sequence horizon to be optimized.
> >Q4.2 & Limitations. Fewer sensors. Ablations: Sensor count vs. horizon.
>
> We thank the reviewer for the question. Our setting is highly sparse: 64–256 sensors cover only 1.56%–6.25% for PDE datasets and 0.08%–0.16% for SST. We further evaluate fewer sensors (<64) and varying $H$ on NS$_{1e-3}$ dataset: increasing $H$ does not consistently help, likely due to a mismatch between training-time reward estimation and test-time execution.
> |Methods|H|32|16|8
> -|-|-|-|-
> PhySense|-|2.75e-3|25.2e-3|0.157
> LASER|1|2.03e-3|7.35e-3|0.116
> ||3|1.95e-3|8.08e-3|0.132
> ||5|1.97e-3|8.03e-3|0.135

---

> > ### Author Rebuttal · Reviewer_xwSg · 2026-04-04
> >
> > Thanks for the insightful rebuttal. After reading the other reviews and rebuttal, I think the paper is in a good shape and should be published. In particular, the iterative training results of world model and policy improve the paper. Happy to increase my score to 4.

---

### Official Review · Reviewer_1x42 · 2026-03-12

**Soundness:** 3
**Presentation:** 2
**Significance:** 3
**Originality:** 3
**Overall Recommendation:** 4
**Confidence:** 4

**Summary:**

This paper proposes LASER, a closed-loop framework for active sensor placement in continuum
physical field reconstruction. The problem is formulated as a POMDP over a learned latent space.
The framework consists of two main components: (1) a continuum field latent world model with a
sparse observation encoder ($p_{\phi}^{\text{enc}}$, cross-attention-based VAE), a latent dynamics
predictor ($p_{\phi}^{\text{dyn}}$, GRU + conditional diffusion), and an implicit neural field decoder
($p_{\phi}^{\text{dec}}$); and (2) a Transformer-based RL policy $\pi_{\theta}$ that outputs continuous
sensor displacements conditioned on predicted future latent states $\hat{z}_{t+1}$ and current
observations $o_t$. The policy is optimized using GRPO with dynamic group filtering and multi-step
predictive reward rollouts. Experiments on Navier-Stokes, Shallow-Water, and real-world Sea Surface
Temperature datasets show improvements over static and offline-optimized sensor placement baselines.

**Compliance With Llm Reviewing Policy:**

Affirmed.

**Final Justification:**

I maintain my score as **Weak accept**. The rebuttal addressed my concerns of the source of gains (active sensing), modest gains over PhySense (choatic dynamics and model capacity), distributional shift in sensing policy improves over random), and finally iterative co-training (consistent improvements). The paper brings **well-known ideas together to form an original contribution in the field and has sound experimental evidence**. Clarity can be improved, with minor suggested changes. Although significant in its field, **broad applicability of the algorithm is limited**.

**Key Questions For Authors:**

1. **Source of gains:** How much of LASER's improvement over baselines is attributable to the
   stronger world model $\phi$ versus the active sensing policy $\pi_{\theta}$? Running PhySense's
   sensor optimization on top of LASER($\phi$) would isolate the active sensing contribution. A
   positive result here would significantly strengthen the paper's central claim.

2. **Modest gains over PhySense:** The improvements on SST (Table 5) and NS$_{\nu\text{1e-5}}$
   are small. Is this a fundamental limitation (diminishing returns of adaptive over well-optimized
   static placement) or an artifact of the decoupled training (frozen $\phi$ during policy learning)?
   Would end-to-end fine-tuning of $\phi$ and $\theta$ help? The answer would clarify the ceiling
   of the active sensing approach.

3. **Metric definitions:** Can you explicitly define Rollout MSE and Reconstruction MSE? Is it
   per-timestep average or cumulative over the trajectory? Is the spatial average over all grid
   points in $\Omega$?

4. **Distributional shift:** How does $p_{\phi}^{\text{dyn}}$'s prediction accuracy change under the
   sensor distributions induced by the learned policy versus the random distributions seen during
   world model training? Quantifying this gap would clarify whether the frozen world model
   introduces compounding errors.

**Limitations:**

The authors acknowledge reliance on world model fidelity and computational overhead, which is
appreciated. However, two additional points deserve discussion: (a) the requirement for ground-truth
fields during RL training (rewards use $u_{t+1}$) and implications for real-world deployment where
ground truth is unavailable, and (b) disentangling the contributions of the world model versus the
policy to understand where future effort should be directed.

**Strengths And Weaknesses:**

### **Strengths**

**S1 (Originality/Significance):** The integration of world model ideas with RL/GRPO for active
sensor placement in continuum field reconstruction is novel. Casting adaptive sensor placement as a
latent POMDP with a learned world model is a natural yet underexplored idea. The paper clearly
motivates why static or offline-optimized layouts (e.g., PhySense) are insufficient for non-stationary
dynamics, and the closed-loop integration of reconstruction, prediction, and sensing is well-justified.

**S2 (Soundness):** The experimental evaluation is thorough within the scope of the paper. Three
datasets of increasing complexity are evaluated, multiple sparsity levels ($N \in \{64, 128, 256\}$)
are tested, and in-distribution (In-t) vs. out-of-distribution (Out-t) temporal segments are separated.
Relevant baselines (AROMA, DiffusionPDE, PhySense) and an RL ablation (LASER$_{\text{PPO}}$)
are included. Ablation studies on history length (Table 6), dynamic filtering (Table 7), and reward
horizon (Table 8) are informative.

**S3 (Originality):** The adaptation of GRPO from LLM alignment to continuous active sensing with
dynamic group filtering is a novel methodological contribution. The "proactive sensing via latent
imagination" idea — conditioning $\pi_{\theta}$ on $\hat{z}_{t+1}$ rather than only $o_t$ — is
well-motivated and empirically validated (Table 8).

**S4 (Soundness):** Sensitivity analyses (Appendix F.1–F.2) demonstrate robustness to initial sensor
configurations, though they are limited to 3 seeds.

### **Weaknesses**

**W1 (Significance — Major):** The gains over PhySense are modest, particularly on the real-world
SST dataset (Table 5: 0.6932 vs. 0.7059 with 100 sensors, 0.7184 vs. 0.7287 with 50 sensors).
On the highly turbulent NS$_{\nu\text{1e-5}}$ benchmark, improvements over PhySense are marginal
across all sparsity levels. It is unclear whether this is a fundamental limitation (diminishing returns
of adaptive vs. well-optimized static placement) or an artifact of the decoupled training pipeline
(world model frozen during policy optimization). End-to-end fine-tuning of $\phi$ and $\theta$ may
close this gap but is not explored.

**W2 (Soundness):** The comparison with PhySense may not be entirely fair. LASER uses a different
and apparently stronger reconstruction backbone — LASER($\phi$) already substantially outperforms
AROMA with random sensors (Table 3). It is unclear how much of LASER's advantage comes from
the better world model $\phi$ versus the active sensing policy $\pi_{\theta}$. Running PhySense's
sensor optimization on top of LASER's world model would isolate the active sensing contribution.

**W3 (Presentation — Minor):** The world model is a core contribution, yet the main paper lacks a
module-level data-flow schematic. Figure 2(a) is a probabilistic graphical model showing conditional
dependencies, not an implementation diagram.

The paper would benefit from a schematic showing:

$o_t \stackrel{p_{\phi}^{\text{enc}}}{\longrightarrow}  z_t \stackrel{p_{\phi}^{\text{dyn}}(\cdot \mid z_t, h_t) (\text{K-step diffusion})}{\longrightarrow} \hat{z}_{t+1} $,

 and for decoder: $z_t \stackrel{p_{\phi}^{\text{dec}}(\cdot \mid x_{\text{grid}})}{\longrightarrow} \hat{u}_t$ during training.

The grey arrow in Figure 3 effectively *is* the world model pipeline but is not labeled as such.
Appendix B provides architectural detail, but a concise data-flow diagram in the main paper is
warranted.

**W4 (Presentation — Minor):** Key definitions and metrics are underspecified:
- The reconstruction loss $\mathcal{L}(u_t, \hat{u}_t)$ is never explicitly defined (presumably MSE
  over all grid points in $\Omega$).
- $-\log p_{\phi}^{\text{dec}}(u_t \mid z_t)$ in Eq. (4) is not stated to reduce to MSE under a
  Gaussian decoder assumption.
- The distinction between "Rollout MSE" (Table 3) and "Reconstruction MSE" (Table 4) should be
  made more precise: Rollout MSE encodes only $o_{t=0}$ and autoregressively predicts via
  $p_{\phi}^{\text{dyn}}$ with no further observations; Reconstruction MSE uses new observations at
  every step. Whether MSE is summed or averaged over time steps should be stated explicitly.
- The flow from sensor positions to loss should be clarified: observations at continuous sensor
  coordinates are obtained via bilinear interpolation (input side), while reconstruction loss is
  evaluated at grid points (output side).

**W5 (Soundness — Minor):** Sensitivity analyses use only 3 random seeds. Some standard deviations
in Table 11 are on the same order as improvements over baselines (e.g., NS$_{\nu\text{1e-3}}$ with
64 obs). Confidence intervals or significance tests in the main tables would strengthen the claims.

---

> ### Author Rebuttal · Authors · 2026-03-31
>
> We thank the reviewer for the valuable comments.
> >W1, Q1, Q2. Modest gains over PhySense on SST and NS$_{1e-5}$. Fine-tuning $\phi$ and $\theta$.
>
> We thank the reviewer for the insightful observation. Regarding improvements on SST and NS$_{1e-5}$, we note that these regimes exhibit highly chaotic dynamics, where reconstruction performance is fundamentally limited **not only by sensor placement but also by model capacity**. The observed diminishing returns of adaptive sensing across sparsity levels suggest that the reconstruction accuracy is constrained by the intrinsic predictability.
>
> To further examine this, as suggested by the reviewer, we conduct **iterative co-training** to mitigate the gap between the world model and policy. By finetuning the world model on policy-collected data and re-optimizing the policy, we observe consistent gains. As shown in the results for SST, **this tighter coupling further enhances LASER's performance over PhySense**. We will incorporate the results and analysis in the revision.
>
> SST|100|50
> -|-|-
> PhySense|0.7059|0.7287
> LASER|0.6932|0.7184
> LASER Co-train|0.6727|0.6934
>
> >W2. PhySense's optimization on LASER's world model.
>
> We need to clarify that the comparison was intentionally designed to isolate the contribution of the **sensing strategy**. As detailed in Line 885, we implement PhySense optimization framework using the **identical world model architecture and pretrained weights** ($\phi$) as LASER. Consequently, the performance gains reported in Tables 4 and 5 are solely attributed to different sensing paradigms. Our results demonstrate that given the same world model, an agent that proactively navigates the field significantly outperforms PhySense’s static layout.
> >W3. Lacks a module-level data-flow schematic.
>
> We appreciate this insightful suggestion. In the revision, we will include a module-level data-flow schematic complementing the probabilistic graphical model, covering both the latent world model and the policy forward pass.
> >W4. Key definitions and metrics underspecified.
>
> We thank the reviewer for pointing out the issues. We will add those clarifications in the revision.
> >W5. Sensitivity analyses use only 3 seeds. Some std in Table 11 are large.
>
> First, we need to correct a typo in Table 11: The standard deviation multiplier for the In-t metrics is $10^{-3}$, not $10^{-2}$. The stochastic variance across seeds is an order of magnitude smaller than previously indicated. To further ensure the robustness of our conclusions, we have expanded our sensitivity analysis from 3 to 5 random seeds. The updated results remain consistent with Table 11 and Table 4, indicating that the observed improvements are stable. We will include confidence intervals in the revision.
> >Q3. Metric definition.
>
> Rollout MSE is the per-timestep average over the trajectory given only $o_{t=0}$, while reconstruction MSE is the per-timestep average with new observations; both are averaged over the spatial domain.
> >Q4. Distributional shift.
>
> The concern that $p^{\text{dyn}}_\phi$ might suffer from compounding errors when moving away from the training (random) distribution is well-taken. We evaluate the world model on policy-induced versus random sensor distributions to quantify this effect.
>
> Specifically, we initialize at $t=1$ using sensor locations selected by the policy (based on $t=0$), and roll out over the full trajectory on NS$_{1e-3}$ dataset with 256 sensors. This is compared against rollouts starting from randomly sampled sensors.
>
> ||In-t($\times 1e-3$)|Out-t($\times 1e-2$)|Avg($\times 1e-2$)
> -|-|-|-
> Random|0.606|0.971|0.525
> Policy Distribution|0.576|0.498|0.284
> Iterative Co-training|0.570|0.408|0.237
>
> Evaluated on the same pretrained $\phi$, the policy-induced distribution improves $p^{dyn}_\phi$ performance, especially in Out-t. This indicates that the policy selects sensor locations that are critical for reconstruction. This suggests that the distribution shift exists but is not harmful. Furthermore, iterative co-training yields additional gains, indicating that tighter coupling between the world model and policy can further enhance performance.
> >Limitations.
>
> **Ground-truth requirement.** We clarify that access to $u_{t+1}$ is required only during RL training to define the reward, not at deployment. In practice, the training phase can leverage high-resolution simulation datasets. At deployment, the learned policy operates solely on observations. (See response to o5Er, W2)
>
> **World model vs policy contributions.** We isolate the contribution of active sensing by comparing different sensor strategies under the same world model in Table 4:
> - random static layouts (AROMA),
> - offline-optimized global layouts (PhySense),
> - sensing heuristics (see responses to o5Er, Q2),
> - adaptive policy.
>
> All models share the same backbone. Results show that adaptive policy consistently outperforms static placements, indicating that the gains are not solely attributable to the world model.

---

> > ### Author Rebuttal · Reviewer_1x42 · 2026-04-03
> >
> > Thank for for addressing the weaknesses. Most of them are resolved. From the sensing standpoint, the algorithms exhibit the the following properties:
> >
> > PhySense: No active sensing, no overhead for sensor placement.
> > LASER: Active sensing, so it requires forward calls to the policy network for each sensor placement.
> >
> > 1. Is my understanding correct?
> > 2. If so, then is the overhead for running the model significant in practice, especially in real-time sensing applications?

---

> > > ### Author Response · Authors · 2026-04-03
> > >
> > > We thank the reviewer for the valuable comments.
> > >
> > > 1. Your assessment is **correct**. The fundamental difference in operational complexity is:
> > >     - PhySense (Static): The sensor layout is determined offline. At inference time, the model only performs a single forward pass of the reconstruction backbone
> > >     - LASER (Active): Each timestep requires a closed-loop cycle:
> > >         - Observe and infer: encode sparse observation and infer predictive latent with $\phi$.
> > >         - Act: forward calls to $\pi_\theta$ to output displacement.
> > >         - Reconstruct: reconstruct the full field with new sparse observations with $\phi$.
> > > 2. Is the overhead "significant" in practice? While LASER is technically more complex, the word "significant" depends entirely on the temporal scale of the physics it is monitoring. At inference time, LASER’s policy and latent prediction happen in a highly compressed latent space. A forward pass of the policy and the latent world model on RTX 3090 typically takes around 16.8ms.
> > >
> > > |Action|Method|Infer/ms|
> > > |-|-|-|
> > > |Place|LASER|2.03|
> > > ||PhySense|-|
> > > |Reconstruct|LASER/PhySense|14.87|

---

### Decision · Program_Chairs · 2026-04-30

**Decision:**

Accept (spotlight)

**Comment:**

This paper considers the high-fidelity measurements of continuum physical fields. This problem remains challenging under sparse and constrained sensing esp vis a vis conventional reconstruction methods' reliance on fixed sensor layouts. In contrast, one can immediately adapt to evolving physical states which would require solving a dynamic decision DP.

The paper then provides two main contributions:
1. It, in effect, proposes LASER, a unified, closed-loop framework that formulates active sensing as a Partially Observable Markov Decision Process (POMDP); and, then,
2. It employs a continuum field latent world model that captures the underlying physical dynamics and provides intrinsic reward feedback which can be used to guide an RL.

Additionally, the proposed RL enables a reinforcement learning policy in order to simulate “what-if” sensing scenarios within a latent imagination space. Furthermore, there are illuminating experiments that demonstrate that LASER consistently outperforms static and offline-optimized strategies.

The rebuttal prepared by the authors was extremely thoughtful and clarified many issues; all reviewers commanded the authors for providing thoughtful response. All, but one reviewer, increase their ratings.